# Understanding the Mechanisms of Chemotherapy-Related Cardiotoxicity Employing hiPSC-Derived Cardiomyocyte Models for Drug Screening and the Identification of Genetic and Epigenetic Variants

**DOI:** 10.3390/ijms26093966

**Published:** 2025-04-23

**Authors:** Abhishikt David Solomon, Swarna Dabral, Raman Gulab Brajesh, Billy W. Day, Matea Juric, Jacek Zielonka, Zeljko J. Bosnjak, Tarun Pant

**Affiliations:** 1Adams School of Dentistry, Oral and Craniofacial Biomedicine, University of North Carolina, Chapel Hill, NC 27599, USA; abhishiktsolomon914@gmail.com; 2Maharishi Markandeshwar College of Pharmacy, Maharishi Markandeshwar (Deemed to be University), Mullana, Ambala 133207, India; swarnadabraljh@gmail.com; 3Department of Biomedical Engineering and Bioinformatics, Swami Vivekanand Technical University, Durg 491107, India; brajeshrg@csvtu.ac.in; 4ReNeuroGen LLC, Milwaukee, WI 53122, USA; billywday@gmail.com; 5Department of Biophysics, Medical College of Wisconsin, 8701 Watertown Plank Road, Milwaukee, WI 53226, USA; mjuric@mcw.edu (M.J.); jzielonk@mcw.edu (J.Z.); 6Department of Medicine, Medical College of Wisconsin, 8701 Watertown Plank Road, Milwaukee, WI 53226, USA; zbosnjak@mcw.edu; 7Department of Surgery, Medical College of Wisconsin, 8701 Watertown Plank Road, Milwaukee, WI 53226, USA

**Keywords:** chemotherapy, cardiotoxicity, hiPSCs, cardiomyocytes

## Abstract

Chemotherapy-related cardiotoxicity (CTRTOX) is a profound and common side effect of cancer-based therapy in a subset of patients. The underlying factors and the associated mechanisms contributing to severe toxicity of the heart among these patients remain unknown. While challenges remain in accessing human subjects and their ventricular cardiomyocytes (CMs), advancements in human induced pluripotent stem cell (hiPSC)-technology-based CM differentiation protocols over the past few decades have paved the path for iPSC-based models of human cardiac diseases. Here, we offer a detailed analysis of the underlying mechanisms of CTRTOX. We also discuss the recent advances in therapeutic strategies in different animal models and clinical trials. Furthermore, we explore the prospects of iPSC-based models for identifying novel functional targets and developing safer chemotherapy regimens for cancer patients that may be beneficial for developing personalized cardioprotectants and their application in clinical practice.

## 1. Introduction

Chemotherapy-related cardiotoxicity (CTRTOX) is a profound and common side effect of therapy in a subset of patients [1,2]. Left ventricular dysfunction (LVD) leading to progressive heart failure (HF) symptoms is the most common short- and long-term complication of chemotherapy-based treatment [3]. Although CTRTOX is causally linked to mitochondrial impairment, increased apoptosis, dysregulated autophagy, increased fibrosis, impaired Ca^2+^ handling, contractile function, and most importantly, oxidative stress due to excess reactive oxygen species (ROS) production, the specific effects and underlying mechanisms of action due to chemotherapy-based treatment are mainly unknown. It is currently impossible to accurately predict which patients will be affected by CTRTOX [4]. Furthermore, recent clinical trials have highlighted the challenges associated with conventional chemotherapy strategies to avert CTRTOX and suggest alternative chemotherapy choices for cancer patients [5].

Due to the practical and ethical barriers associated with accessing human subjects and their ventricular cardiomyocytes (CMs), most studies on CTRTOX have been carried out using animal models [6,7,8]. Although these models have improved our understanding of the pathological mechanisms of CTRTOX, their inability to accurately recapitulate the human disease phenotype, physiology, and progression due to differences in species’ backgrounds, genetics/epigenetics, and developmental patterns has decelerated reliable clinical translation in human clinical trials [9,10]. Additionally, discrepancies in the cardiac parameters such as anatomy, kinetics, contractility, and action potential have led to substantial discordance between animal models and are a significant hurdle in clinical translation from bench to bedside [9]. The advent of stem cell technology and regenerative medicine in the past decade has paved new paths in experimental approaches, allowing researchers to use stem cell types (e.g., embryonic, induced, and adult) for modeling cardiac diseases.

Stem cells are undifferentiated biological cells found in multicellular organisms with the potential to show sustained proliferation and differentiate into other cell types [11]. For example, embryonic stem cells (ESCs) have been widely explored with a determined aim to generate functional cardiomyocytes (CMs) and rejuvenate the damaged myocardium [12,13]. ESCs are pluripotent stem cells derived from the inner cell mass of the blastocyst, maintaining their undifferentiated state under appropriate conditions [14,15]. Once provided the stimulus, they can differentiate into specific cell types like muscle, nerve, blood, and heart cells [16,17]. In the present scenario, having the whole heart transplanted as the only feasible option with the limited number of donors imposes a significant challenge [18]. Human ESCs, because of their robust proliferation capacity and ability to differentiate to form functional CMs, are an ideal candidate for repairing the myocardium. Despite the recent results obtained in a few of the in vivo studies where they depict a partial ability to revascularize the damaged myocardium, significant challenges like the generation of high-purity CMs, ethical issues related to embryo accessibility, immune rejection, teratoma formation, and the need for a proper delivery method still limit the clinical application of ESCs [19,20]. Overall, human ESCs appear promising for generating cardiac disease models to study and provide better insight into disease development and pathogenesis. Still, substantial challenges, such as limited availability, tissue rejection, carcinogenic risk, host incompatibility, immunogenicity, and ethical concerns, restrict their clinical application.

Having a relatively limited differentiation potential compared to ESCs, mesenchymal stem cells (MSCs) have been widely explored to generate functional CMs and rejuvenate the damaged myocardium. MSCs can be obtained from bone marrow, umbilical cord, placenta, and adipose tissue [21,22]. MSCs give rise to osteocytes, chondrocytes, adipocytes, and cardiomyocytes. When MSCs were administered, CMs and endothelial cells were reinforced in a mouse heart attack model [23]. MSCs are therapeutically advantageous due to their immunomodulatory effects and can secrete soluble growth factors and cytokines in a paracrine fashion [24,25,26,27]. Even though MSCs have been shown to improve cardiac function, poor cell retention and inadequate differentiation hinder their application in cardiac regenerative therapy. After the transplantation of MSCs, a therapeutic effect was not observed, likely due to increased immune cell retention and infiltration, which resulted in the rejection of MSCs [28,29].

The unique ability of induced pluripotent stem cells (iPSCs) to efficiently differentiate into different cell types of the myocardium with no associated ethical and immunogenicity concerns provides unparalleled opportunities to understand the mechanisms of cardiovascular disease (CVD) pathogenesis and develop new therapeutic strategies with the potential to treat human diseases.

In 2006, Takahashi and Yamanaka et al. demonstrated that iPSCs can be derived from adult cell reprogramming in the presence of transcription factors like octamer-binding transcription factor 4 (Oct 4), SRY (sex determining region Y)-box 2 (Sox-2), myc proto-oncogene c-Myc, and Kruppel-like factor 4 (Klf-4) [30]. Since then, human iPSCs have been generated via invasive approaches using fibroblasts or vein endothelial cells and non-invasive techniques using blood and urine cells [30,31,32,33]. In the past few years, published studies from different research groups, including ours, have shown successful recapitulation of the genetic/epigenetic background by iPSCs’ differentiated cardiac cells from patients with CVDs [34,35,36]. Additionally, these induced pluripotent stem-cell-derived CMs (iPSC-CMs) have been extremely useful in understanding the mechanistic foundation and progression of cardiac diseases, allowing us to generate environmentally and genetically driven robust in vitro models in human cells for modeling cardiac diseases [36,37]. Moreover, these patient-specific iPSC models have successfully captured the complex pathological interactions and phenotypic surrogates of CVDs, allowing the discovery and testing of therapeutic strategies with ever-increasing clinical significance [38,39].

The heart is an essential organ comprising a range of cell types that can broadly be divided into two subgroups: cardiomyocytes (atrial, ventricular, and nodal) crucial for cardiac function and non-cardiomyocytes (fibroblasts, smooth muscle cells, pericytes, and epicardial, endocardial, and endothelial cells) providing structural, vascular, and mechanical support to the CMs. Intriguingly, all these heart cells have been generated using iPSCs in two-dimensional (2D) monolayer cultures [36,40]. The emergence of three-dimensional (3D) organoids and advances in cutting-edge technologies, such as gene editing and single-cell RNA sequencing, have brought unprecedented opportunities for modeling CVDs and deciphering the mechanisms relevant to heart development and diseases in vitro.

This review aims to present the current knowledge about the underlying mechanisms of CTRTOX and the methods to counteract it. Additionally, we explore the innovative use of human induced pluripotent stem-cell-derived CMs (hiPSC-CMs) to study CTRTOX and their role in identifying genetic and epigenetic variants to understand patients’ sensitivity to chemotherapeutic drugs. Finally, we also discuss the applications of hiPSCs-CMs in screening drug efficacy and precision medicine for heart diseases.

## 2. CTRTOX Pathophysiology and Mechanisms

Over decades, CTRTOX has emerged as a serious concern, as the chemotherapeutic agents have been associated with the potential to cause cardiovascular complications and severe toxicity to the heart (Figure 1), while the molecular mechanisms of the interplay between chemotherapy drugs and cardiotoxicity remain mostly unknown. This section explicitly discusses the current understanding of the pathophysiological mechanisms associated with CTRTOX. Table 1 summarizes the effects and mechanisms of the distinct chemotherapy strategies from the preclinical and clinical studies 7 (21%) involved patients; 11 involved cell models (33%).

### 2.1. Anthracycline-Induced Cardiotoxicity

Anthracyclines are antineoplastic antibiotics encompassing many different drugs, including doxorubicin, daunorubicin, epirubicin, idarubicin, and mitoxantrone [73]. They continue to be essential to the therapy against solid and hematological tumors. Regardless of chemotherapeutic action, the cumulative dose-dependent cardiotoxicity-induced senescence phenotype in distinct cell types of the heart and other significant consequences such as a decline in left ventricular ejection fraction (LVEF), cardiomyopathy, and irreversible heart failure significantly limit the clinical application of anthracyclines [74,75,76,77]. Over the years, studies with experimental models have shown that the cardiotoxicity mediated by anthracyclines is attributed to several mechanisms, including oxidative stress, DNA damage, calcium dysregulation, mitochondrial dysfunction, inflammation, and apoptosis [78,79,80,81]. However, the underlying mechanisms associated with anthracycline-induced cardiotoxicity (AIC) are not entirely known and remain to be deciphered. For instance, cardiac anthracycline toxicity is attributed to the drug’s ability to cause severe DNA damage primarily by inhibiting topoisomerase IIβ (Topo IIβ), thereby inducing double-stranded DNA breaks, thus resulting in the activation of the p53 tumor suppressor pathway to increase apoptosis in cardiac cells [41,82]. Besides this, other classical mechanisms via which anthracyclines damage the myocardium are linked to mitochondrial dysfunction, which might lead to the increased generation of free radicals and oxidative-stress-induced apoptosis. Anthracyclines are well known to bind to the phospholipid cardiolipin [42,83]. Several studies point out that this electrostatic binding can promote mitochondrial dysfunction by disrupting the optimal enzymatic activity of cardiolipin-dependent mitochondrial electron transport chain (ETC) complexes I, III, and IV [43,44,81]. Of note, anthracycline accumulation within the mitochondria leads to increased free radical production, including ROS and reactive nitrogen species (RNS), inducing lipid peroxidation and mitochondrial DNA (mtDNA) damage and inhibiting ETC and causing decreases in intracellular adenosine triphosphate (ATP) level, resulting in apoptosis in the CMs [43,44,84]. For instance, anthracyclines were found to reduce the production of nitric oxide (NO) while increasing superoxide radical anion (O_2_^•−^) production via modulating endothelial nitric oxide synthase (eNOS) transcript and protein expression, as well as directly interacting with the protein, to induce oxidative stress, subsequently resulting in the generation of cellular oxidants including peroxynitrite (ONOO^−^) and hydrogen peroxide (H_2_O_2_) [45,46,47,48,49].

NAD(P)H dehydrogenase-mediated reduction of anthracyclines forms a semiquinone radical that generates O_2_^•−^ in reaction with molecular oxygen, resulting in the redox cycling process and increased H_2_O_2_ production [85]. H_2_O_2_ may further react with transition metal ions, including Fe^2+^ and Cu^+^, giving rise to a hydroxyl radical (^•^OH) to irreversibly damage biomolecules [86]. Overall, doxorubicin (DOX)-induced oxidative stress imparts diverse effects in cells such as interfering with nitric oxide synthase (NOS) and NADPH oxidase (NOX) activities, increasing iron uptake, decreasing nuclear factor erythroid 2-related factor 2 (Nrf2), superoxide dismutase (SOD), cytochrome c (Cytc), and adenylate kinase 4 (AK4) proteins via modulating the expression of protein tyrosine phosphatase 1B (PTP1B), and the transactivation of p53 and upregulation of BCL2-associated X (BAX) leading to cardiotoxicity [45,50,51]. Additional mechanisms such as DOX-induced reduction of AMP-activated protein kinase (AMPK) pathways and the subsequent increased activation of stress-induced AKT serine/threonine kinase (Akt) and mitogen-activated protein kinase (MAPK) in the heart were associated with DNA damage, enhanced stress, and the hypertrophy of cardiac tissues [52]. Furthermore, evidence of AIC via influencing mitochondrial biogenesis, oxidative mitochondrial metabolism, apoptosis through the inhibition of the mammalian target of rapamycin (mTOR) signaling, autophagy through ULK1, and decreased fibrosis through the inhibition of TGFβ signaling has also been reported [87]. Of note, a considerable amount of research shows anthracyclines can trigger severe inflammatory responses, promoting adverse cardiac events, eventually leading to LVD and, in extreme cases, congestive heart failure. In the past, many animal models have substantiated that co-administering antioxidants could normalize oxidative stress to mitigate the cardiovascular toxicity of anthracycline [88,89,90]. Despite extensive research, our understanding of the internal targets and biochemical mechanisms of AIC remains limited and warrants further investigation. This section emphasizes the anthracycline drugs in clinical use and the best-known signaling mechanisms underlying the cardiotoxicity associated with them.

#### 2.1.1. Doxorubicin

DOX is a toxic anthracycline agent obtained from Streptomyces spp. [91]. It is widely used to treat leukemias and solid tumors in pediatric and adult patients [92]. Its action involves Topo IIβ inhibition via DNA intercalation, resulting in DNA damage, thereby reducing tumor cell mobility in rapidly proliferating tumors [93]. Although for decades DOX has been among the most effective and widely used anticancer drugs, commonly used to treat both child and adult cancer types, its use is limited by its most common and well-established cardiotoxic side effect: cardiomyopathy leading to HF in a subset of patients. LVD leading to progressive HF symptoms is the most common short- and long-term complication of DOX exposure, occurring in up to 10% of patients. DOX cardiotoxicity is cumulative-dose-dependent, and the increased risk of CVD begins with the first dose of chemotherapy. In general, there is a markedly higher incidence of clinically significant cardiomyopathy with a cumulative dosage of 450 to 500 mg/m^2^, which is an ideal dose for treatment. Additionally, several studies thus far have substantiated that DOX modulates several cardiac cytochrome P450 (P450) enzymes, including cytochrome P450 3A4 (CYP3A4) and cytochrome P450 2D6 (CYP2D6), with subsequent alterations in P450-mediated cardiotoxic and cardioprotective pathways [94,95].

#### 2.1.2. Daunorubicin

Daunorubicin (DNR) is another anthracycline, with a mechanism of action similar to DOX, widely used in cancer chemotherapy [96]. In recent years, several studies have shown that DNR administration resulted in cardiotoxicity [53]. For instance, Arozal and colleagues reported that DNR at a cumulative dose of 9 mg/kg induced cardiac damage among rats treated with 3 mg/kg body weight DNR intravenously for a week. Cardiac damage presented in interstitial edema, subendocardial fibrosis, perinuclear vacuolation, and disorganization and degeneration of the myocardium [54]. A recent case study demonstrated that induction chemotherapy with DNR (60 mg/m^2^) for AML treatment in a 46-year-old woman was found to be associated with cardiotoxic side effects, including a reduction in the left ventricular systolic function, a decline in the ejection fraction of >10%, and an increase in cardiac troponin. In another in vivo study, intraperitoneal (IP) DNR administration at a dose of 6 × 3 mg/kg every 48 h and 15 mg/kg was shown to induce subchronic and acute cardiomyopathy, specifically in the 10–12-weeks male Wistar rats, characterized by a reduction in left ventricular mass and function and increased expression of natriuretic peptides in both subchronic and acute cardiomyopathy [55]. Additionally, the cardiotoxic effect in subchronic cardiomyopathy was attributed to DNR-mediated increased oxidative stress and the decreased expression of chemoattractant cytokines and cardiac regenerating stem cells [55]. Functional and mechanistic studies on the implication of DNR administration negatively impacting cardiac function are also emerging. For example, in a rat model of subacute DNR (3 mg/kg, intravenous; every 48 h) and subchronic DNR (15 mg/kg; intravenous), DNR-associated cardiomyopathy (DACM) reduction in left ventricular function and LV weight was accompanied by the upregulation of natriuretic peptides and a decrease in Myh6 to total Myh ratio [56]. Additionally, the administration of DNR, particularly in the subchronic model, dysregulated the myocardial micro RNA (miR-208a, miR-499, miR-1, and miR-133a) network, promoting the late progression of DACM [56]. Intriguingly, in recent years, studies were examining the regulation of intracellular calcium levels and its role in the development of DNR cardiomyopathy. Kucerova et al. demonstrated that six IP doses of DNR (3 mg/kg, every 48 h) in male Wistar rats were associated with depressed left ventricular function, an upregulation of calcium release channel ryanodine receptor type 2 (RyR2), a concomitant increase in natriuretic peptide precursor A (NPPA) and natriuretic peptide precursor B (NPPB), and a decrease in alpha-tubulin gene expressions indicating a particular role of RyR2 in DACM [56]. Also, an in vitro investigation using rat H9c2 cardio myoblast (H9c2) cardiomyocytes demonstrated that DNR markedly induced apoptosis by modulating the PI3K/Akt signaling pathway via the regulation of Ca^2+^ influx, suggesting the role of the DNR-sensitized calcium channel in cardiac dysfunction [57].

#### 2.1.3. Epirubicin

Epirubicin (EPI) is another anthracycline Topo IIβ inhibitor clinically used as an adjuvant medication with cancer medications to treat cancer patients [97]. Many animal studies involving the administration of EPI have demonstrated cardiotoxic events. For example, Guven et al. observed that a single dose of EPI, 10 mg/kg IP in Wistar rats, increased mitochondrial degeneration, swelling, intracytoplasmic vacuolization, and focal myofilament disarray in the cardiomyocyte and subsequently induced cardiotoxicity [58]. Another study found that in male Sprague-Dawley rats, the administration of EPI (8 mg/kg, IP injection) induced cardiotoxicity via the upregulation of genes promoting autophagy and apoptosis [59]. Like animal studies, the effect of EPI-based chemotherapy in patients with aggressive lymphomas has also demonstrated a substantial risk to heart functions. In a clinical trial, an increase in the electrical properties of the heart, including QT dispersion and QTc dispersion, was observed among non-Hodgkin lymphoma patients administered intravenous EPI (a bolus dose of 40 mg/m^2^) for 3 weeks compared to the control [60]. In a phase II open, non-randomized trial, a cumulative EPI dose of 200 mg/m^2^ among cancer patients was found to induce impairment in systolic left ventricular (LV) function characterized by a reduction in the strain rate (SR) peak at 3, 6, 12, and 18 months of follow-up that correlated with the expression of inflammatory interleukin-6 (IL-6) and oxidative stress markers [61]. Peng et al. also reported that EPI at a dose of 100 mg/m^2^ upregulated apoptosis while downregulating 5′-aminolevulinate synthase 2 (ALAS2) related to “glycine, serine, and threonine metabolism” in a subset of breast cancer patients, suggesting the role of these metabolic pathways in the development of symptomatic cardiomyopathy after EPI-based chemotherapy [62].

#### 2.1.4. Idarubicin

Idarubicin (4-demethoxydaunorubicin) is a newer anthracycline used to treat solid tumors, lymphomas, and leukemias in adults and children [98]. While, in a clinical setting, the cases of idarubicin-related cardiotoxicity are less common, idarubicin administration seems to cause subclinical myocardial dysfunction, a decrease in LVEF, and myelodysplasia (MDS) and cardiomyopathy in acute myeloid leukemia (AML) patients. Yang et al. demonstrated that a patient with AML with no risk factors of heart disease developed severe subacute congestive heart failure (CHF) and ventricular tachycardia (VT) in response to her first induction chemotherapy involving a total cumulative idarubicin dose of 36 mg/m^2^ [63]. The idarubicin treatment prolonged the cardiac QTc interval and frequent premature ventricular contractions with a QTc interval of 400 ms [63]. In another study, it was found that idarubicin treatment induced cardiac arrest in a patient diagnosed with AML [64]. A recent case report described that a woman without any cardiac risk factors treated for AML developed idarubicin cardiomyopathy with the first exposure to cumulative dosages of 36 mg/m^2^, specifically within 2 weeks after initiating idarubicin chemotherapy. This is supported by the fact that idarubicin chemotherapy reduced LVEF by 25%, right ventricular (RV) function, and induced severe mitral regurgitation, thus inducing severe cardiomyopathy [65].

#### 2.1.5. Mitoxantrone

Mitoxantrone (MTX) is a substituted 1,4-dihydroxy-9,10-anthraquinone that is a clinically relevant chemotherapy medication with antitumor activity used to treat advanced stages of tumors, including leukemia, lymphoma, and breast cancer [66]. Compared to other anthracycline regimens, the usage of MTX was found to be significantly less cardiotoxic among cancer patients. Regarding the cardiotoxic effect of MTX, Rossato et al. demonstrated that MTX alone and combined with its naphthoquinoxaline metabolites induced CYP450- and CYP2E1-mediated cytotoxicity in H9c2 cells. This cardiotoxic effect of MTX was also observed in vivo, where MTX and naphthoquinoxaline metabolites accumulated in the heart and liver of MTX-treated rats [67]. Another study demonstrated that 24 h of exposure to MTX (5 µM) disrupted energetic pathways in 7-day differentiated H9c2 cells to induce autophagy-induced toxicity [68]. Costa et al. noted that clinically relevant concentrations of MTX (1 and 10 µM) were found to alter proteomic, energetic, and oxidative stress homeostasis in HL-1 cardiomyocytes [69]. Additionally, evidence from in vivo studies in adult and infant mice demonstrated that the MTX (7.0 mg/kg) treatment of adult mice induced myocardial injury and fibrosis, increased NF-κB p52 and tumor necrosis factor alpha (TNF-α), while decreasing IL-6 inflammatory gene expression, triggering cardiotoxicity [70]. This cardiotoxic effect of MTX was also observed in MTX (6 mg/kg)-treated adult CD-1 male mice via modulating cardiac metabolism, decreasing glycolysis, and increasing the dependency on fatty acid (FA) oxidation, namely, through decreased AMP-activated protein kinase (AMPK) and glyceraldehyde-3-phosphate dehydrogenase (GAPDH) content. MTX decreased free carnitine (C0) and increased acetyl carnitine (C2) concentrations [71]. A recent case study described that a patient with AML undergoing salvage chemotherapy with MTX developed acute myocarditis [72]. Notably, it was reported that MTX reduced LVEF by 25% and caused diffuse myocardial edema and delayed gadolinium enhancement [72].

In conclusion, based on the data generated from numerous preclinical and clinical studies, AIC is a significant concern that can lead to severe cardiovascular complications. Henceforth, efforts are warranted to identify circulating surrogate biomarkers for the risk prediction of AIC, patient-specific dose management, and the use of modified delivery approaches.

### 2.2. Targeted-Therapy-Induced Cardiotoxicity

Targeted therapy is a type of cancer treatment that includes drugs or different substances targeting specific mechanisms and proteins in cancer cells. The class includes drugs like trastuzumab, lapatinib, sunitinib, gefitinib, afatinib, sorafenib, erlotinib, pazopanib, tucatinib, and neratinib, which are widely utilized either as a single-drug therapy or in combination with other treatments for treating various malignancies, particularly non-small cell lung cancer (NSCLC). Despite their effectiveness, these drugs have been associated with varying degrees of cardiotoxicity.

#### 2.2.1. Trastuzumab

Trastuzumab (TZM) is an effective antineoplastic agent used in various cancers like human epidermal growth factor receptor 2 (HER2)-positive breast cancer and metastatic gastric cancers [99]. Its mechanism of action is via targeting Her-2/erbb2, which is essential for cardioprotection and cardiac development. Reports have suggested that its administration either alone or combined with anthracyclines causes impairment in cardiac progenitor cells, development in microvascular networks, and heart failure [100,101,102]. Nevertheless, the cardiotoxicity mediated by TZM is reversible [103]. It decreases LVEF, causing left ventricular systolic dysfunction, and induces structural alteration in heart tissues [101,104,105,106]. Treatment with TZM also elevates serum triglycerides (TGs), very low-density lipoprotein (VDL), cardiac troponin I (cTnI), lactate dehydrogenase (LDH), caspase-3, caspase-9, and malondialdehyde (MDA) levels. It decreases BCL-2, CAT, SOD, GPx, and GST levels, suggesting oxidative stress and apoptosis, finally resulting in cardiomyocyte congestion and microthrombus development in the coronary arteries [107,108,109]. Since TZM blocks HER2 as its mechanism of action, it causes cardiotoxic events most likely due to its on-target effect. It not only compromises the cardioprotective effects via blocking neuregulin/HER2/erBB2 signaling, but it also alters calcium signaling and mitochondrial dysfunction, activates the Erk/mTOR/Ulk 1 cascade resulting in the potentiation of oxidative stress in heart and endothelial cells, inhibits autophagy, and induces cardiomyocyte death either alone or when combined with anthracyclines [110,111,112,113,114,115,116]. Mitochondrial dysfunction and oxidative stress in cardiomyocytes also manifest due to the inhibition of the translocation of cardioprotective mitochondrial Cx43, resulting in cardiac cell apoptosis [117]. Apart from on-target cardiotoxic events, TZM mediates cardiac outcomes via its off-target activity on Topo IIβ, downregulating post-TZB administration and resulting in cardiac fibrosis and damage [110,118,119]. It also upregulates pro-apoptotic protein, i.e., BCL-xS, and downregulates STAT3, causing mitochondrial dysfunction, oxidative stress, and cardiomyocyte death. TZM also interferes with PI3-Akt signaling and the focal adhesion kinase (FAK) cascade and downregulates the neuregulin-1 survival signaling pathway, which synergizes the damage due to oxidative stress in the heart. TZM interferes with the RAAS system via the upregulation of angiotensin II (ANG II), resulting in increased oxidative stress in the heart and endothelial cells, endothelial dysfunction, and cardiomyocyte apoptosis [120,121]. Microarray studies have revealed that the administration of TZM resulted in an alteration in the expression of 15 genes associated with cardiac and mitochondrial functionalities, stress responses, and DNA repair, which eventually results in the alteration of cardiac structure, impairment of LVEF, and cardiotoxic outcomes [106].

#### 2.2.2. Lapatinib

Lapatinib (LAP) is a quinazoline derivative and a small tyrosine kinase inhibitor against HER1, HER2, and EGFR [122]. Patients administered with LAP were observed with manifestations of adverse cardiac effects and a reversible decrease in LVEF [123,124]. However, reports have revealed that LAP’s cardiotoxic potential is less severe than TZM [125]. It impairs Erbb2-Erbb4 signaling and activates NRG-1, decreasing MAPK phosphorylation and Akt in cardiac tissues. Moreover, Erbb2 inhibition increases mitochondrial ROS generation, which causes cardiotoxicity [126]. Additionally, when combined with DOX, LAP increases the accumulation of DOX in cardiac cells, activates PI3-Akt signaling, and increases iNOS, resulting in mitochondrial dysfunction, the elevation of DOX-mediated oxidative stress, ferroptosis, and apoptosis in cardiomyocytes [127,128,129,130]. In brief, the cardiotoxicity associated with LAP is primarily due to its inhibitory action on cardiac ErbB2-ErbB4 signaling, which activates NRG-1 and decreases MAPK phosphorylation and Akt. This leads to increased mitochondrial ROS production, potentiating its cardiotoxicity.

#### 2.2.3. Sunitinib

Sunitinib (SUN) is an oral multitargeted tyrosine kinase inhibitor (TKI), including vascular endothelial growth factor receptors (VEGFRs), that has been proven effective in metastatic renal cell carcinoma (mRCC) [131]. However, patients with mRCC treated with SUN reported a modest decline in LVEF, hypertension, and heart failure [132,133,134,135,136]. Fms-like tyrosine kinase 1 (sFlt-1), a ligand of VEGF, is associated with the severity of heart failure. This demonstrates the protective effect of the VEGF cascade for cardiovascular health, and the inhibition of this pathway via SUN is one mechanism contributing to its toxic cardiovascular potential [137,138,139]. The inhibition of VEGF signaling also leads to the elevation of endothelin 1 in the circulation, which eventually leads to prolonged vasoconstriction, hypertension, and cardiac outcomes [140]. Since SUN is a multiple kinase inhibitor, reports have suggested that its inhibition of FGF2 is also a potential reason for its cardiotoxic potential [141]. Besides VEGF signaling, SUN also inhibits PDGFR and decreases the coating of pericytes in the vessels, resulting in leaky pericytes and cardiac dysfunction [142]. It increases interleukin-1β (IL-1β), interleukin-6 (IL-6), interleukin-8 (IL-8), and interleukin-18 (IL-18) abundance and miR-133a expression, inhibits NF-κB, and decreases phosphorylation of the ASK1/MKK7/JNK pathway in cardiomyocytes, causing cardiac inflammation and fibrosis [143,144,145,146]. SUN also results in cardiac hypertrophy by upregulating MAPK and AhR/CYP1A1 pathways [147,148]. Treatment with SUN results in a dose-dependent decrease in PI3K activity, the generation of ROS, a decrease in the shortening of sarcomeres, and a decrease in the calcium transient in the myocardium, resulting in a negative ionotropic effect and cardiac manifestations [149]. SUN increases MDA content, Fe^2+^, and TfR expression levels while decreasing glutathione (GSH), Nrf2, and GXP4 expression, supporting that SUN demonstrates a cardiotoxic effect via oxidative stress and Nrf2-dependent ferroptosis pathways [150]. It increases pPKCα levels, inhibits ribosomal S6 kinase (RSK), and results in the activation of pro-apoptotic factor BCL2-antagonist of cell death (BAD), resulting in the activation of BAX and release of cytochrome c, subsequently initiating apoptosis and depleting ATP. Furthermore, it also inhibits the AMPK/mTOR/autophagy pathway, resulting in ATP depletion and hypoxia, which results in myocyte loss, LV impairment, and hypertrophy [151,152,153,154,155]. SUN impairs the cardiomyocyte survival mediator cellular communication network factor 2 (CCN2), resulting in cardiac dysfunction [156]. It also promotes calcium/calmodulin-dependent protein kinase IIδ (CaMKIIδ) expression and CaMKII, which leads to cardiac toxicity [157].

#### 2.2.4. Gefitinib

Gefitinib (GEF) is a selective erbb1 receptor antagonist linked with cardiotoxicity; like ERL, its cardiotoxic profile is relatively less severe than other antineoplastic drugs [158,159]. Its mechanism of cardiotoxicity includes the inhibition of cardiac EGFR and reduction in Akt levels, followed by an increased expression of phosphatase and tensin homolog (PTEN) and forkhead box O3 (FoxO3) genes, culminating in hypertrophy, apoptosis, and elevated plasma troponin levels. Additionally, GEF is metabolized by CYPYA1 in cardiac microsomes into active metabolites, which further cause cardiac injury [160]. It also induces the angiotensin II type 1 receptor (AT1R) and angiotensin II receptor type 2 (AT2R), leading to overactivation of the RAAS and AngII/AT1R/NOX pathway, resulting in the elevation of oxidative stress, induction of the MAPK pathway, and cardiac hypertrophy [161]. Furthermore, GEF upregulates BNP and β-MHC while it downregulates α-MHC mRNA, indicating oxidative stress, apoptosis, and cardiomyocyte hypertrophy. Moreover, GEF inactivates hERG and partially blocks KCNQ1/KCNE1, resulting in delayed repolarization and prolonged QT intervals [162,163,164].

#### 2.2.5. Afatinib

Afatinib (AFA) is another EGFR and HER2 tyrosine kinase inhibitor used for NSCLC. A few clinical reports have chiefly inferred that AFA holds a safer cardiac profile with either no or minimal reported cardiotoxicity compared to other antineoplastics [165,166]. Nevertheless, it can stimulate endoplasmic reticulum (ER) stress and amplify the expression of inflammatory genes like NF-κB, IL-6, and IL-1β, exacerbating inflammation and cardiac events [167]. These reports suggest that the cardiac manifestations associated with AFA are less pronounced, and the long-term effect of AFA on cardiac health warrants more research.

#### 2.2.6. Sorafenib

Sorafenib (SOR) is approved for the treatment of solid tumors like renal cell carcinoma (RCC) and has some reported cardiovascular outcomes [133]. It reduces heart size and induces myocyte necrosis and hypertrophy [168]. SOR, being a tyrosine kinase inhibitor, antagonizes MS-like tyrosine kinase (FLT-3), VEGFR and platelet-derived growth factor receptor-beta (PDGFR), Raf-1 and extracellular signal-regulated kinase (ERK), which interferes with angiogenesis, triggers apoptosis, and influences cardiomyocyte survival [169,170]. It also reduces Slc7a11 and GPX4, leading to ferroptosis, oxidative stress, mitochondrial damage, and cardiotoxicity [171]. SOR also downregulates cardioprotective stanniocalcin 1 (stc1), thus causing cardiomyocytes to be more vulnerable to SOR-induced oxidative and endoplasmic stress, resulting in cardiac inflammation [167,172]. It alters RMB20 and downregulates SLC25A3 and FHOD3, thereby impairing the ATP synthesis and survival of mitochondria, eventually leading to cardiac toxicity [173]. SOR reduces phospholamban phosphorylation; increases intracellular calcium levels, mitochondrial oxidative stress, and mitochondrial damage; activates CaMKII; and reduces mitochondrial complex II, leading to myocardial fibrosis, impaired autophagy, hypertrophy, and apoptosis [174,175,176,177,178]. SOR causes cardiac inflammation as evidenced by increased TNF-α, IL-1β, IL-6, NLRP3, and TLR4. It causes cardiac apoptosis, as observed by elevated caspase-3, TGF beta, and BCL2 expression in cardiomyocytes [179].

#### 2.2.7. Erlotinib

Erlotinib (ERL) is an EGFR inhibitor used to treat non-small lung cancer (NSLC). Although it has a relatively lower cardiotoxic profile than other anticancer drugs, there have been some reports of cardiomyopathy and acute cardiovascular events in a few patients post-ERL administration. Mechanistically, ERL-induced cardiac events are associated with decreased plasma magnesium levels, the activation of neutrophils, elevated levels of substance P, and increased oxidative stress [180,181,182,183]. Although these manifestations are considerable, several studies have reported that ERL is relatively safer and presents lower apoptotic effects in cardiomyocytes than other antineoplastic drugs [184,185].

#### 2.2.8. Pazopanib

Pazopanib (PAZ) is used for multiple neoplasms; however, it is also associated with cardiotoxicity as observed via hypertension, prolongation of the QT interval, left ventricular strain, and heart failure. One plausible mechanism is the inherent property of PAZ to inhibit VEGF2, which can eventually increase endothelin1 in the circulation, causing the prolongation of vasoconstriction as observed via SUN [140,186]. Likewise, VEGF-associated microvascular proliferation and cardiomyocyte survival can also be impacted by PAZ administration. Justice et al. have explicitly described how the inhibition of VEGF by PAZ can contribute to various cardiotoxic events.

In conclusion, targeted therapies are effective anticancer drugs with varying degrees of cardiotoxic profile, and more investigations are required to deduce the mechanisms of toxicity and the strategies to mitigate them.

### 2.3. Macrolide-Induced Cardiotoxicity

Macrolides encompass a crucial class of orally active antibiotics, including mitomycin, erythromycin, and clarithromycin, that have demonstrated the potential to inhibit the growth of several tumor and cancer cell lines [187,188,189,190]. Although highly effective, the clinical use of these drugs has been associated with adverse cardiovascular events such as myocardial infarction (MI), prolongation of the QT interval, delayed rectifier potassium current (IKr)-induced ventricular tachyarrhythmia, and sudden cardiac death (SCD) [191,192,193]. In past years, preclinical research substantiated that the biological pathways associated with macrolide-induced cardiac events include oxidative stress, DNA damage, mitochondrial malfunctions, the release of cytochrome c, and apoptosis. Considerable research efforts are still required to identify the specific targets and biochemical basis of macrolide-induced cardiotoxicity (MIC). This section will review the well-established signaling mechanisms and physiological changes typically related to MIC.

#### 2.3.1. Mitomycin C

Mitomycin C (MMC) is a bio-reductive alkylating agent used as a chemotherapeutic to treat various types of cancer, including breast cancer, adenocarcinomas of the GI tract, and others [194,195,196]. The precise mechanisms of MMC-induced cardiotoxicity are yet to be deciphered. However, oxidative stress, DNA damage, mitochondrial dysfunction, and endothelial dysfunction are the primarily reported mechanisms associated with MMC-induced cardiotoxicity. In preclinical studies, pathological cardiac changes have been observed in rats treated with MMC derivatives, including BMY-26605, BMY-25282, BMY-25551, and BMY-25690 [197]. Additionally, clinical trials have indicated the incidence of heart failure (from 2.2 to 10%) among patients treated with MMC (30 mg/m^2^) in combination with DOX (range 100–800 mg/m^2^). Another cohort study (*n* = 180) demonstrated that 14 of 91 (15.4%) patients treated with MMC and before DOX developed symptomatic heart failure, suggesting there could be a synergistic effect of MMC and DOX involved in declining heart function [198].

#### 2.3.2. Erythromycin and Clarithromycin

Erythromycin (ERT) and clarithromycin (CLT) are the most extensively used macrolides. However, both are implicated in severe cardiotoxic effects. These compounds destabilize the cardiac autonomic nervous system, resulting in LVD and congestive heart failure. These antibiotics upregulate ERG1 gene expression, exacerbating cardiac toxicity. The molecular mechanisms underlying the cardiac manifestations via ERT and CLT are not entirely understood but include oxidative damage, mitochondrial impairment, and apoptosis, which contribute to the destabilization of the cardiac autonomic nervous system, culminating in catastrophic cardiac events. A study has reported that ERT induces oxidative stress by increasing MDA levels and decreasing superoxide dismutase (SOD) levels, leading to apoptosis marked by elevated levels of p53, Bcl-2, Bax, caspase-3, and caspase-9 in the heart. Another study demonstrated that ERT impairs sodium influx and cardiac cell depolarization via downregulating the sodium voltage-gated channel alpha subunit 5 (Nav1.5) transcript, further contributing to cardiac fibrosis. Similarly, clarithromycin has been reported to cause ventricular dysrhythmias or ventricular tachycardia. Both erythromycin and clarithromycin promote the formation of ROS, mitochondrial swelling, and cytochrome c release in cardiomyocytes, leading to arrhythmia, QT prolongation, and Torsade’s de Pointes.

While macrolides have demonstrated promising efficacy when used in combination with chemotherapeutics such as antineoplastic agents, their potential to cause cardiac manifestations via oxidative stress, mitochondrial damage, and the impairment of cardiac ion channels necessitates continued research into the mechanisms for mitigating these events.

## 3. Recent Advancements in Therapeutic Strategies in the Treatment of CTRTOX: Success and Limitations

The above-described studies clearly indicate that while the chemotherapy regimens attenuate cancer progression, these have not been without their limitation of CTRTOX in cancer patients. In this section, we address the challenges that encompass issues such as the implications of chemotherapy in cancer patients and different approaches to prevent and treat CTRTOX and further discuss the cardiac safety of these approaches (Figure 2 and Table 2).

### 3.1. Pharmacological Drug Selection

Recent studies have suggested varying degrees of heterogeneity among patients after exposure to anthracyclines. For instance, some patients exposed to higher doses of anthracyclines exhibit minor cardiac anomalies compared to their corresponding groups, developing catastrophic cardiac dysfunction or potentially deadly congestive HF despite being exposed to lower DOX levels. Reducing the chemotherapeutic dose and prioritizing drugs with a lesser cardiotoxic profile is one strategy to avoid cardiac manifestations due to anthracycline administration. In recent years, cardiovascular agents, including β-adrenergic antagonists, carvedilol, and nebivolol, have been found to exert cardioprotective and vasodilatory benefits in a subset of cancer patients undergoing chemotherapy. Carvedilol is a third-generation FDA-approved β-blocker with vasodilatory and antioxidant action that has been established as an efficient drug to improve cardiovascular complications among patients with chronic heart conditions. The cardioprotective effect of carvedilol has been investigated in clinical trials with recently diagnosed cancer patients undergoing anthracycline-based chemotherapy. For example, in a randomized controlled trial with breast cancer patients, an intake of 6.25 mg carvedilol daily during chemotherapy was found to mitigate the cardiac strain and strain rate, suggesting that carvedilol-based prophylaxis can prevent DOX-induced cardiotoxicity [199]. Nabati et al., in a randomized, single-blind, placebo-controlled study, demonstrated that the prophylactic use of carvedilol among recently diagnosed breast cancer patients inhibited AIC as evidenced by reduced troponin I levels, intact LVEF, and increased left ventricular end-systolic volume (LVES) and LA diameter in carvedilol-treated patients (*n* = 46) compared to the placebo (*n* = 45) [200]. A prospective, randomized, double-blind, placebo-controlled study showed that among 200 randomized patients with a HER2-negative breast cancer tumor status and normal LVEF referred for ANT (240 mg/m^2^), the use of carvedilol resulted in a significant reduction in troponin levels and diastolic dysfunction [201]. Also recently, Carrasco et al. conducted a pilot, randomized controlled, two-arm clinical trial to assess the effect of carvedilol administration, as well as non-hypoxic myocardial preconditioning based on docosahexaenoic acid (DHA), in breast cancer patients undergoing anthracycline treatment [202]. In preclinical and clinical studies, nebivolol is another third-generation β-adrenergic blocker reported to exert a protective effect against anthracycline-induced cardiotoxicity. Imbaby et al. demonstrated that the oral administration of nebivolol (1 and 2 mg/kg) improved heart index, cardiac enzymes, histopathological features, and ECG parameters, reducing DIC in rats [203]. In a prospective double-blinded study, the prophylactic treatment of a breast cancer patient (*n* = 27) with nebivolol (5 mg) daily protected the myocardium against AIC. Notably, the left ventricular (LV) end-systolic and end-diastolic diameters remained unchanged at the baseline and the end of 6-month chemotherapy [204]. Overall, these studies suggest that compared to control subjects, carvedilol- and nebivolol-treated patients exhibited better heart function, and thus the drugs should be considered as a preferential chemotherapy regimen in high-risk patients with baseline cardiovascular complications or receiving cumulative anthracycline doses.

Preclinical findings from recent studies have demonstrated dexrazoxane as a possible precautionary strategy. Dexrazoxane is a prodrug counterpart of the metal chelator ethylenediaminetetraacetic acid (EDTA) and may remove iron from the iron–doxorubicin complex, resulting in decreased ROS formation [205]. According to a more recent study, dexrazoxane may have a cardioprotective effect because it specifically degrades Topo IIβ, a protein implicated in DIC, whereas it ignores Topo IIα in human leukocytes, which is essential for DOX anticancer action [220]. This recent finding creates opportunities for novel approaches, including administering dexrazoxane first and then doxorubicin later, thus preserving DOX efficacy. Several research studies, including systematic reviews and meta-analyses, have shown its effectiveness in lowering cardiovascular risk, cardiac abnormalities, and cardiotoxicity symptoms. Dexrazoxane has been demonstrated to reduce half the risk compared to control groups, minimize the possibility of cardiac dysfunction by threefold, and reduce cardiotoxicity effects by 74% [206]. However, only one study demonstrated a modest protective effect against the loss of LVEF, indicating that more research is necessary to fully understand its usefulness in cardioprotection from CTRTOX.

Among the central mechanisms, oxidative stress is one of the most frequently encountered phenomena in CTRTOX. An interesting and promising strategy could be to target mitochondria to mitigate oxidative stress. The mitochondrial-targeted antioxidants (MTAs) provide a novel approach for addressing oxidative stress at its origin by explicitly targeting mitochondria [221]. In recent years, evidence from preclinical studies using MTAs, including mitoquinone (Mito-Q) and Mito-Tempol, has demonstrated that they can increase antitumor effectiveness and lessen the cardiotoxicity linked to chemotherapeutic drugs. A critical work by Chandran et al. suggests a novel cardioprotection mechanism of Mito-Q in DOX-treated rats via restoring cytochrome c oxidase (CcO subunits II) activity in the heart tissue [81]. In another study, Dickey et al. (2013) examined the combined effects of DOX, Mito-Tempol, and dexrazoxane using a breast cancer model of a syngeneic rat [207]. The findings showed that Mito-Tempol and dexrazoxane successfully decreased DIC without sacrificing doxorubicin’s anticancer properties. Increased autophagy, a cellular mechanism that promotes survival, reduced apoptosis, or programmed cell death in the heart, has been associated with the cardioprotective effect [207]. The study suggests a unique function for protein oxidation markers in cardiovascular protection by identifying serum proteins that undergo oxidation under cardiotoxic conditions. This preclinical research offers compelling justification for investigating Mito-Tempol’s potential in therapeutic settings [207].

In a different study, Kim et al. (2020) examined the use of Mito-FF, a mitochondria-targeting self-assembly peptide, in conjunction with 5-fluorouracil (5-FU) to treat gastric cancer [208]. With reduced antioxidant enzyme activity and increased production of ROS in gastric cancer cells, their results demonstrated the value of the combination treatment as compared to a monotherapy. Crucially, the combo therapy’s pro-apoptotic effects were eliminated when ROS generation was inhibited, suggesting that increased oxidative stress is the primary mechanism [208]. Although the emphasis of this study was gastric cancer, the results lend credence to the more general idea that by modifying ROS levels, mitochondria-targeted medicines can improve anticancer effects.

### 3.2. Delivery Strategies and Targeted Therapies

The advent of nanotechnology in cancer treatment has made for promising strategies such as the liposome-mediated delivery of chemotherapy regimens, providing several advantages over conventional chemotherapy, including (1) the reduced exposure of healthy tissues, (2) reduced toxicity, (3) targeted delivery, and (4) controlled release. Studies suggest that advancing these therapies in larger patient populations and clinical trials in cancer treatment may be beneficial. An advanced approach of using liposome-based nanocarriers to ensure effective targeted drug delivery and prevent their biodegradation has significantly improved their stability at the target site and helped reduce chemotherapy-induced cardiotoxicity. Numerous liposomal formulations are currently designed to increase DOX’s therapeutic index [222,223]. These formulations change the pharmacokinetics and biodistribution of DOX by encasing it in a lipid bilayer [224]. PEGylated liposomal DOX (PLD), such as Doxil (Caelyx globally), and non-PEGylated liposomal DOX are the most widely used liposomal DOX formulations [225,226]. By decreasing the reticuloendothelial system’s (RES) absorption of the liposomes, PEGylation—the polyethylene glycol (PEG) binding on the liposome surface—increases the liposomes’ circulation time [209,210,211]. While lowering the contact with healthy tissues, such as the heart, this extended circulation permits a more significant buildup of DOX in cancerous tissues [209,211].

A study utilizing the FDA Adverse Event Reported System (FAERS) database concluded that liposomal DOX formulations had lower reported odds ratios (RORs) for myelosuppression, cardiotoxicity, and alopecia than standard DOX [226]. PEGylated formulations of liposomal DOX demonstrated a greater ROR for palmar–plantar erythrodysesthesia (PPE) [226]. Several randomized trials have reported that compared to conventional chemotherapy, PEGylated liposomal-based administration for the first treatment of different cancer types has shown comparable efficacy and reduced cardiotoxicity. In 2004, a randomized clinical trial with metastatic breast cancer patients investigated the effectiveness of CAELYX (PEGylated liposomal DOX) to DOX and found that compared to 60 mg/m^2^ DOX (every 3 weeks), the administration of 500 mg/m^2^ of PEGylated liposomal DOX (every 4 weeks) resulted in significantly reduced cardiotoxicity, myelosuppression, vomiting, and alopecia in the breast cancer patients [212]. Similarly, Huang et al., in a clinical study involving breast cancer patients, observed that the sequential use of PEGylated liposomal doxorubicin (PLD) with trastuzumab mitigated clinical cardiotoxicity, demonstrated as asymptomatic decreased LVEF, compared with the results obtained in previous clinical studies using conventional anthracycline, taxanes, and trastuzumab [213]. In another randomized clinical trial, the use of liposomal DNR (80 mg/m^2^ per day for 3 days) in AML patients demonstrated antileukemic activity comparable to idarubicin (12 mg/m^2^ per day for 3 days) and induced subclinical or mild cardiomyopathy and was well tolerated in patients [214]. Intriguingly, a group of researchers also identified that attaching a NO-releasing moiety to a PEGylated derivative of EPI (p-EPI-NO) further increased cytotoxicity against Caco-2 cancer cells while decreased toxicity against non-neoplastic cells, including H9c2 and adult cardiomyocytes [214]. Santucci et al. demonstrated that compared to conventional EPI, PEGylated EPI, BP-747-treated mice were devoid of cardiotoxicity and displayed significantly reduced tumor volume [215]. Likewise, the administration of MTX in anionic cardiolipin liposomes increased the therapeutic index of MTX and resulted in negligible cardiotoxicity [216]. Additionally, the administration of liposomal resveratrol (LIPO-RES) at a dose of 20 mg/kg for 6 weeks in rats conferred protection against DOX-induced oxidative stress, inflammation, and calcium dysregulation [217]. According to a different study conducted on domestic pigs, liposomal DOX was less cardiotoxic than conventional DOX, showing a significant decrease in myocardial fibrosis and improving both left and right ventricular systolic functioning [227]. This decrease in cardiotoxicity is explained by the activation of interferon-related DNA damage resistance and decreased myocardial drug accumulation [227]. Compared to intravenous DOX and intravenous LipoDox, a study employing an innovative polyelectrolyte-stabilized liposome (layersome) production reported lower cardiotoxicity [228]. This was demonstrated by more significant amounts of GSH and SOD and lower levels of creatine phosphokinase (CK-MB), LDH, and MDA [228]. Additionally, it is important to note that not all studies have reliably shown that liposomal DOX formulations minimize cardiotoxicity, with some reporting that cardiotoxicity can be comparable or even higher under certain situations or with specific formulations [224]. This demonstrates the intricacy of the connection and the necessity of more investigations to completely comprehend the variables affecting cardiotoxicity in various liposomal DOX formulations. Apart from the studies mentioned above, more recently, a group of researchers identified that the duodenal administration of idarubicin-loaded solid lipid nanoparticles to rats reduced drug uptake in the heart [218].

Another strategy for preventing CTRTOX can be tumor-targeted therapies. For instance, by selectively targeting hormone-sensitive tumors that overexpress specific receptors on cancer cells, researchers hope to reduce the collateral damage to the heart while enhancing the effectiveness of cancer treatments. As an example, gonadotropin-releasing hormone (GnRH) receptor-based conjugates represent an innovative and promising strategy for targeted drug delivery in cancer treatment. Polgar et al., using primary human cardiac myocytes (HCMs) and human umbilical vein endothelial cells (HUVECs) in vitro, demonstrated that GnRH conjugates containing DOX, DNR, and methotrexate had no cytotoxic effect on HCMs [219]. As research progresses, these conjugates could become part of a new generation of precision therapies, particularly for hormone-sensitive cancers or tumors that overexpress GnRH receptors. However, future clinical applications must address receptor heterogeneity, linker stability, and resistance challenges.

In short, the approaches mentioned above may be promising for treating different cancers. However, at present, the field faces several practical challenges, including (a) the inability to translate animal work to study CTRTOX, (b) the optimization of chemotherapeutic doses in animals as seen in humans, (c) inter-patient variability in cancer-treatment-related cardiotoxicity among cancer survivors, (d) unidentified genetic and epigenetic loci predisposing patients to CTRTOX, and (e) limited access to human ventricular CMs. Henceforth, it is necessary to use a high-throughput method to generate human CMs. With advances in the application of human induced pluripotent stem cells (hiPSCs) to generate CMs, sequencing technologies, and genome editing techniques, a few of the challenges mentioned above can be addressed in human cell culture models. In the next section, we provide a comprehensive overview of the application of hiPSC-CMs to examine patient-specific responses to chemotherapy treatments and predict the factors associated with treatment-related cardiotoxicity among cancer survivors.

## 4. Application of Human iPSC-Derived Cardiomyocytes to Study CTRTOX

The human heart is a terminally differentiated organ comprising approximately 2–3 billion CMs. These specialized cardiac cells are about 100 μm long and 10–25 μm in diameter, primarily building up the muscle walls (myocardium) of both the chambers (atria and ventricle), constituting a significant proportion of the heart. The CMs, the primary cell types, are also involved in contractile function, allowing the heart to pump out blood, and have also been shown to be affected in numerous CVDs, leading to contractile dysfunction. A significant challenge in cardiac research has been accessing the primary tissues of healthy and diseased patients to study changes in the specific cell types during cardiac development and disease. Therefore, over the past few years, multiple protocols have been developed to culture and differentiate hiPSCs to CMs in 2D monolayers and 3D organoids to understand better the developmental and cell-specific changes in CMs during cardiac development and disease (Figure 3). Using 2D and 3D CM models from hiPSCs provides an innovative way to study and develop heart cell differentiation and illness. These models could be pivotal in creating safer, more effective cancer treatments and precision cardiology. Table 3 summarizes the experimental procedures modeling CTRTOX using hiPSC-CMs.

### 4.1. Patient-Specific iPSC-Based 2D and 3D CM Models: Platform to Investigate CTRTOX and Identify Genetic and Epigenetic Variants

hiPSC-CMs represent a powerful platform for identifying the molecular pathways of CTRTOX. In recent years, examining patient-specific responses to chemotherapeutic drugs using hiPSC-CMs has allowed researchers to gain better insights into the mechanisms underlying CTRTOX, identify individuals at higher risk, and test potential therapeutic strategies to develop personalized cardioprotectants. For example, Burridge et al. demonstrated that patient-specific hiPSC-CMs can recapitulate the predilection to DIC of individual patients at the cellular level. hiPSC-CMs derived from individuals with breast cancer who experienced DIC recapitulated the increased sensitivity to DOX toxicity than hiPSC-CMs from patients who did not experience CTRTOX [229]. In another study, DOX treatment (>6 µM) in vitro was shown to exert a cardiotoxic effect on hiPSC-CMs via altering mitochondrial calcium levels and membrane potential, subsequently modulating mitochondrial function and cell viability [230]. Umesh et al. also reported that repeated hiPSC-CM exposure to DOX leads to mitochondrial dysfunction and energy depletion. Utilizing NMR-based metabolic profiling, the authors also investigated the metabolic consequences of hiPSC-CM exposure to DOX and concluded that acetate and formate could be valuable biomarkers for DOX-induced toxicity, with implications for predicting long-term cardiovascular damage and assessing drug safety in preclinical models [231]. Similarly, to better replicate the chronic cardiotoxicity observed in cancer patients undergoing chemotherapy, Karhu et al. showed that the low-dose exposure of hiPSC-CMs to DOX (100 nM) over a long period (up to 21 days) induced cell death, reduced cell viability, and DNA damage in hiPSC-CMs, with increased caspase-3/7 activity indicating apoptosis [232].

Besides the anthracyclines mentioned above, other investigators have assessed other classes of anticancer drugs also being tested on in vitro models of 2D and 3D CMs derived from hiPSCs. For example, Doherty et al., using a multiparameter toxicity screen, investigated the cardiotoxic potential of FDA-approved tyrosine kinase inhibitors (TKIs), including crizotinib, SUN, and nilotinib, and reported the distinct mechanisms underlying these toxicities [182]. Overall, the authors demonstrated the varying cardiac toxicity profiles of different TKIs, with some, like SUN, crizotinib, and nilotinib, having more pronounced adverse effects on hiPSC-CM health. At the same time, ERL appears less harmful to cardiac function [182]. Sharma et al. found that hiPSC-CMs from cancer patients (*n* = 11) who have developed tyrosine kinase inhibitor (TKI)-induced cardiotoxicity recapitulate that increased risk in vitro when exposed to TKIs. In addition to hiPSC-CMs, the researchers also derived hiPSC-derived endothelial cells (hiPSC-ECs) and cardiac fibroblasts (hiPSC-CFs) to study the impact of TKIs on different cardiac cell types. The high-throughput screening revealed that TKIs targeting vascular endothelial growth factor receptor 2 (VEGFR2) and platelet-derived growth factor receptor (PDGFR) were particularly cardiotoxic across all cell types tested (hiPSC-CMs, hiPSC-ECs, and hiPSC-CFs). This suggests that the heart may respond to VEGFR2/PDGFR signaling disruption by activating protective mechanisms to maintain cellular integrity [233]. This highlights the importance of considering the specific cardiac risks when choosing TKIs for clinical use. Additionally, Kurokawa et al. demonstrated that TZM-induced cardiac dysfunction is primarily driven by disruptions in cellular energy metabolism rather than by cell death and that targeting this pathway through AMPK activation could offer a novel approach to mitigate these side effects in cancer patients [234]. Similarly, Kitani et al., using patient-specific iPSC-CMs, demonstrated that TZM-induced cardiac dysfunction primarily results from alterations in energy metabolism rather than cardiomyocyte death, opening the door for further investigation into AMPK activators as potential therapies for cancer patients experiencing TZM-induced cardiac dysfunction [235].

Overall, there is strong experimental evidence that patient-specific iPSC-derived 2D and 3D cardiomyocyte models are useful experimental tools to study CTRTOX and offer a high-throughput, personalized, and mechanistically rich approach to understand and mitigate the cardiovascular risks associated with cancer therapies. In the future, customized 2D and 3D CM models could predict a patient’s risk of developing cardiotoxicity when undergoing specific cancer therapies, including but not limited to chemotherapy, guiding clinicians to tailor treatments or choose cardioprotective strategies for high-risk patients. In addition, CRISPR-Cas9 technology can be employed in iPSCs to introduce genetic variants linked to cardiac vulnerability, helping elucidate how specific mutations or polymorphisms may predispose patients to CTR.

### 4.2. Application of hiPSC-CMs for Identification of the Genetic and Epigenetic Markers of CTRTOX

In recent years, the developed methodologies of generating 2D and 3D CMs from hiPSCs, in combination with transcriptome profiling and scRNA-seq techniques, has provided an advanced platform for evaluating CTRTOX at both the functional and molecular levels (Figure 4 and Table 4). In hiPSC-CMs, treatment with DOX (156 nM) for 2–6 days induced arrhythmic beating and reduced contractile force (amplitude), consistent with disturbances in cardiac function. The apoptosis-related genes (e.g., BAX, FAS) and stress markers (e.g., GPX1, ZMAT3) were upregulated, correlating with human heart failure conditions, suggesting these as potential biomarkers for the early detection of cardiotoxicity [236]. Further, DOX exposure (156 nM) of hiPSC-CMs caused early deregulation of a few micro RNAs (e.g., miR-187-3p, miR-182-5p, miR-34a-3p) that were linked to cardiac dysfunction and heart failure in both patient and animal models [237]. In the assessment of the role of the glutathione S-transferase mu-1 GSTM1 null genotype and AIC, Singh et al. reported a significant link between the GSTM1 null genotype and anthracycline-related cardiomyopathy in childhood cancer survivors. This was supported by evidence of downregulated GSTM1 expression in peripheral blood and in hiPSC-CMs of survivors with cardiomyopathy [238]. Magdy et al. highlight the potential of hiPSC-CMs to study patient-specific differences in DIC, particularly in response to DOX. Utilizing hiPSC-CMs, they demonstrated how genetic variants identified through genome-wide association studies (GWASs), such as the rs2229774 SNP in retinoic acid receptor-γ (RARG), can be validated using this platform to explore the mechanisms underlying DIC and inform personalized treatment strategies [239]. Using hiPSC-CMs models, Magdy et al. also suggested that rs11140490 in the *SLC28A3* locus was linked to reduced DIC. The protective effect was mediated by regulating an antisense long noncoding RNA (SLC28A3-AS1) that overlaps with SLC28A3 [240]. In addition, the implication of RARG-S427L in impairing the DNA repair mechanisms in DIC has also been confirmed in hiPSC-CMs [241]. Their role in DIC was confirmed using CRISPR-Cas9-genome-edited hiPSC-CMs. The RARG-S427L variant resulted in increased DNA damage and more significant mitochondrial reduction in response to DOX exposure compared to RARG-WT/WT cells, suggesting that RARG-S427L may be a pharmacogenetic biomarker for identifying individuals at increased risk of DIC. Using hiPSC-CM cell lines with either intrinsic polymorphism or CRISPR-Cas9-mediated deletion of the *rs28714259* locus, Wu et al. identified a common germline SNP, the rs28714259 risk allele, in a glucocorticoid receptor (GR) binding site that disrupts GR binding at the locus, reducing GR-mediated protective signaling and subsequently diminishing the beneficial effects of dexamethasone (a glucocorticoid) pretreatment on cardiomyocyte survival and contractility [242].

Recent advances in epigenome-wide association studies (EWASs) have made identifying regions with DNA methylation variations linked to various disease phenotypes easier. In a recent study, Singh et al. explored the methylome of peripheral blood at a single-CpG resolution in childhood cancer survivors exposed to anthracyclines. They compared those who developed cardiomyopathy to those who did not. The study identified ten genes with hypermethylated differentially methylated positions (DMPs) and twelve genes with hypomethylated DMPs, specifically in childhood cancer survivors who experienced anthracycline-induced cardiomyopathy [243]. Additionally, knockout experiments in the human induced pluripotent stem-cell-derived cardiomyocytes (hiPSC-CMs) of four of these genes—EXOC6B, FCHSD2, NIPAL2, and SYNPO2—demonstrated an increased sensitivity to DOX, suggesting that these genes may play a protective role for cardiomyocytes against AIC [244]. Singh et al. also investigated the blood-based mRNA expression profiles in childhood cancer survivors exposed to anthracycline chemotherapy, focusing on identifying the molecular markers associated with anthracycline-induced cardiomyopathy [245]. The study used a matched case–control design involving 40 childhood cancer survivors who developed cardiomyopathy (cases) and 64 survivors without cardiomyopathy (controls). LDHA (lactate dehydrogenase A) and CD36 (cluster of differentiation 36) were significantly upregulated in the cases, pointing to altered metabolic processes in the failing heart. IL1R1, IL1R2 (interleukin-1 receptors), and MMP8 and MMP9 (matrix metalloproteinases) were found in multiple canonical pathways, suggesting a role in inflammation and structural remodeling of the heart. The knockout of LDHA in human induced pluripotent stem-cell-derived cardiomyocytes increased sensitivity to DOX, supporting its role in cardiotoxicity. This research highlights the complexity of cardiotoxicity in childhood cancer survivors. Metabolic and inflammatory pathways may be critical targets for further investigation in cardiomyopathy related to anthracycline exposure [245].

In conclusion, hiPSC-CMs provide a robust platform for assessing the cardiotoxic potential of compounds in preclinical testing. Additionally, this platform provides an opportunity to decipher the functional role of genetic and epigenetic variants and develop targeted therapies to activate protective pathways, which could improve outcomes for patients receiving anticancer drug treatments. Further research is needed to refine these strategies and modulate and re-balance the impact of chemotherapeutics on the heart and tumors.

## 5. Challenges and Future Outlook of Utilizing hiPSC-CMs in Studies on CTRTOX

Proof-of-concept studies using hiPSC-CMs to recapitulate the development of cardiotoxicity in cancer patients after treatment with anthracycline or small-molecule tyrosine kinase inhibitors have ushered in the promise of hiPSC-based disease modeling for toxicity screening. Because these studies were retrospectively conducted on patient samples collected after a review of medical charts, such experimental designs must be cautiously interpreted and extrapolated due to the limitations of selection bias. Notwithstanding, substantial barriers to improving the identification of patient-specific approaches exist as unmet needs of precision-guided medicine for cardio-oncology practice. For example, routine toxicity testing has not been prospectively validated, and patient-specific approaches using hiPSC-CMs are at present challenging, cost-prohibitive, and time-consuming. Reprogramming and the maintenance of hiPSCs takes a considerable amount of time, bears a high cost, and is very labor intensive, significantly limiting its potential to be of immediate diagnostic value. Notwithstanding, such proof-of-concept studies represent a significant milestone and usher in a new era for personalized toxicity screening. To overcome these hurdles, the broad goal of future research projects should aim to develop independent, personalized, and predictive diagnostic biomarkers for patient susceptibility to CTRTOX (i), particularly within at-risk cancer groups during the initiation of chemotherapy, and (ii) to validate the utility of this novel intervention for CTRTOX pathology using patient-specific models. Moreover, to maintain the highest possible scientific rigor, the experimental approaches must be carefully designed to ensure a robust and unbiased experimental design, methodology, analysis, interpretation, data integrity, and reporting of the results. Biological variables shall be carefully controlled to ensure reproducible experimental results. Experimental treatments should be randomized, and experimental operators should be blinded to the hypothesis being tested whenever possible.

Our recent efforts have focused on using patient-specific PBMCs/neutrophils to prospectively assess patient tissue-specific effects of high-risk therapeutic regimens and compare the results with patient outcomes after receiving chemotherapy for breast cancer. As we recently reported, the PBMCs/neutrophils derived from breast cancer patients who experienced clinical DIC were more sensitive to doxorubicin toxicity, depicting (1) an increased response to DOX with neutrophils being more sensitive to DOX at lower concentrations than PBMCs; (2) decreased ATP production indicating lower cellular viability; and (3) lower OCR associated with basal mitochondrial respiration, ATP production, maximal respiration, and spare respiratory capacity (representing mitochondrial function), as compared to breast cancer patients without cardiotoxicity [246]. Henceforth, this approach has the potential to prospectively predict the impairment of cardiac function as a surrogate marker for CTRTOX and rapidly and predictably translate into clinical decision making.

Overall, we underscore the application of hiPSC-based 2D and 3D CM models in drug screening, precision medicine, and the modeling of a wide range of CTRTOX cases. However, assessing pharmacologic intervention meaningfully and creating precise therapeutic strategies in the coming era of precision medicine requires multidimensional future research and well-optimized experimental models. Additionally, the following areas require focused attention to enhance the translational potential of hiPSC-derived 2D and 3D CM models.

### 5.1. High-Throughput Manufacturing of 2D and 3D CMs from hiPSCs

The dependence on intricate and expensive matrices makes it challenging to scale up procedures and employ the control and monitoring of conventional 2D cultures for hiPSC proliferation and cardiac differentiation. Nevertheless, these difficulties are successfully addressed by matrix-free 3D suspension cultures that use stirred platforms, like spinner flasks and controlled stirred tank bioreactors (STBRs). This method’s adaptability and effectiveness have been demonstrated by its successful application for hiPSC proliferation and differentiation into various cell types, like CMs, macrophages, endothelium, and its derivatives, proving its versatility and efficiency [247,248,249]. Numerous types of CVDs and drug-induced cardiotoxicities have been extensively modeled using hiPSC-CMs grown on 2D substrates [250,251]. Despite advancements, complex cell–cell and cell–matrix interactions throughout the body are not well captured by 2D hiPSC-CMs [252,253]. Modeling capabilities are currently strengthened by the enhanced replication of in vivo structures made possible by 3D cell cultures [254,255]. 3D hiPSC-CM culture techniques provide increased troponin expression and greater cardiac lineage specificity [256]. Although there is uncertainty over distinguishing qualities, human cardiac organoids generally share several essential traits in common, such as the capacity to evolve as culture time increases, structural complexity akin to tissue-level function, and the ability to replicate native cardiac cellular compositions by using different cell types. Superior biological imitation of native cardiac tissue shape and function has also been made possible by human 3D cardiac organoids, which more precisely mimic heart development and various cardiac diseases [257,258]. Using hiPSC-CMs possesses great potential for drug development, tissue engineering, regenerative medicine, in vitro disease modeling, and other fields. Nevertheless, this will require the regular manufacturing of hiPSC-CMs in sizable amounts customized for specific uses [249].

### 5.2. Improvement of hiPSC-CM Models

The ability of hiPSC-CMs to mimic CTRTOX, such as prolonged arrhythmic beating and cytotoxicity following repeated DOX exposure, has been reported in studies where genes linked to ion homeostasis, apoptotic control, and sarcomere structure were found by transcriptomic analysis to be possible biomarkers for drug screening and safety evaluations. Notably, these reports highlight the shortcomings of conventional models given that, because of variations in experimental time, delayed toxic effects were shown in hiPSC-CMs but not in primary newborn rat ventricular myocytes. The significance of hiPSC-CM-based models of cardiotoxicity in identifying genetic and therapeutic targets can be significantly improved by incorporating physiological mechanical stress, a primary driver in numerous cardiac pathologies. Better models that include mechanical stress may recapitulate in vivo cardiac pathology more accurately, offering a strong foundation for research on cardioprotection and drug development. Adding physiologically relevant mechanical stress to hiPSC-CM models to more accurately emulate the exposure of CMs to mechanical forces occurring in vivo can further increase their usefulness and aid in mechanistic studies and in developing new treatments. Though they frequently lack the maturity and complexity of genuine human cardiomyocytes, current hiPSC-CM models are excellent at mimicking patient-specific electrophysiological traits and arrhythmia syndromes.

### 5.3. Exploration of 3D hiPSC-CMs in Studying the Mechanisms of CTRTOX

CVD brought on by CTRTOX stands as the second leading cause of death for cancer survivors [259]. Effective in vitro models are required to forecast possible cardiotoxicity and evaluate medications for reducing cardiotoxicity during and after treatment [260]. Since the creation of cardiac microtissues and 3D organoids grown from human induced pluripotent stem cells (hiPSCs), new opportunities in cardio-oncology have been made possible. The use of heart-on-a-chip devices made from hiPSCs has revealed the essential pathways involved in the mechanisms of CTRTOX [259]. By combining human cells, microfabrication methods, and sophisticated sensors, heart-on-a-chip systems are created to mimic the composition and operation of a human heart in a microfluidic system. To create more effective interventions for cardiovascular illness, they indicate the potential for personalized care, simulation of disease, and drug screening [261]. Moreover, cardiac microtissues hold promise in developing more efficient strategies to eliminate CTRTOX that involve changes in endothelial cells, cardiac fibroblasts, and hiPSC-CMs. A recent study by Archer et al., 2018, highlights the development and analysis of 3D cardiac microtissues made from hiPSC-generated cardiomyocytes, where the crucial interactions between the cardiomyocytes and non-cardiomyocytes influencing cardiac physiology are captured by these 3D models, thereby combining both the morphological and functional features of the heart [262]. These microtissues offer mechanistic insights into structural cardiotoxicity by utilizing viability assays with high-content biology approaches, such as mitochondrial and endoplasmic reticulum integrity assessments [262,263]. By facilitating the early identification of phenotypic fingerprints and cardiotoxic liabilities, this cutting-edge technology holds promise in developing safer therapeutic drugs and lowering the cardiovascular risks connected to anticancer treatments. This overview outlines the many forms of cardiotoxicity caused by anticancer drugs and emphasizes the clinical and translational difficulties related to the cardiotoxicity of organoids, heart-on-a-chip devices, and cardiac microtissues produced from hiPSCs.

### 5.4. High-Throughput Screening of Genetic, Epigenetic, and Proteome Markers Using Patient-Specific hiPSC-CMs

Using hiPSC-CMs provides an innovative way to study heart cell differentiation and illness and develop new therapeutics. Researchers can decipher the molecular pathways behind CTRTOX by analyzing the genetic, epigenetic, and protein expression profiles of CMs derived from the iPSCs of healthy individuals and cancer patients with and without cardiomyopathy exposed to chemotherapy drugs. For example, the results of many functional studies utilizing hiPSC-CMs have reflected that genetic and epigenetic variants might affect the vulnerability to DIC; similarly, in a DOX model of CTRTOX, a high-throughput analysis of hiPSC-CMs enabled the discovery of important physiological and protein biomarkers linked to cardiotoxicity [243]. Henceforth, a better understanding of the molecular mechanisms underlying CTRTOX can be obtained by examining the metabolic and protein profiles of hiPSC-CMs exposed to distinct chemotherapeutic drugs. This will enable the identification of critical pathways and possible therapeutic targets, especially by capturing the more intricate cellular interactions that occur in a 3D environment that may be overlooked in 2D cultures. Furthermore, the evaluation of the impact of chemotherapy drugs on cardiac electrical signaling and rhythm in a setting that closely resembles the physiological settings can be rendered possible by designing an electrophysiological study utilizing hiPSC-CMs.

### 5.5. Utilizing hiPSC-CMs to Assess Cardioprotective Strategies

Among many drugs, metformin is being proposed to ameliorate DIC [264,265]. It has been a generally accepted notion that metformin use in T2D patients with cancer is beneficial due to its potential antineoplastic and cardioprotective effects [266,267]. The cardioprotective effect of metformin is mediated via the activation of AMPK and has a direct impact on mitochondrial function [268,269]. In animal models and in vitro studies, metformin has been demonstrated to prevent DOX-induced cell death in CMs by reducing ROS and attenuating mPTP opening [270]. Similar effects were observed using SGLT2 inhibitors (SGLT2is). Large-scale clinical studies have shown substantial cardiovascular advantages, including a decreased risk of being hospitalized for heart failure or death from cardiovascular disease, even though these drugs were first created for their glycemic management qualities [271]. These advantages go beyond people with type 2 diabetes; they are effective in heart failure patients of all glycemic levels [272]. Notably, a recent study has demonstrated the cardiac benefits of SGLT2i inhibitor-based therapy in animal models of AIC by preserving myocardial energetics [273]. Henceforth, utilizing the hiPSC-CM cardioprotective responsiveness of metformin and SGLT2i in CTRTOX and the metabolic and signaling pathways involved in the cardioprotective effects are warranted.

### 5.6. Therapeutic Strategies for Restoring Mitochondrial Health in CTRTOX Using hiPSC-CMs

Mitochondrial damage is an essential contributor to CTRTOX. The biosynthesis and function of mitochondria are masterly regulated by PGC-1α [274]. It stimulates mitochondrial biogenesis, lowers oxidative stress, and improves mitochondrial respiration. PGC-1α activation has been demonstrated to enhance mitochondrial function, lower ROS generation, and prevent cardiomyocyte death [275]. The cardioprotective actions of PGC1α are also linked to the Nrf2 pathway, an essential regulator of antioxidant defense. By triggering the Nrf2 pathway, berberine, for instance, has been demonstrated to mitigate doxorubicin-induced cardiac injury and fibrosis by reducing oxidative stress and damage to mitochondria [276]. In recent years, our research group, along with others in the field, has successfully generated hiPSC-CMs for disease modeling and drug assessment. Recently reported PGC-1α overexpression was found to recapitulate the pharmacological prevention by carvedilol of DOX-induced mitochondrial dysfunction in hiPSC-CMs [277]. Henceforth, PGC-1α activation may lessen anthracycline-induced mitochondrial damage, providing possible treatment options for CTRTOX.

### 5.7. Isogenic Cardiac Organoid Models

Characterizing appropriate genetic variations requires the development of isogenic disease models that vary primarily in the DNA mutants of interest. This procedure has been transformed by CRISPR-Cas9 gene editing, which uses clustered regularly interspaced short palindromic repeats [278]. Additionally, iPSC-based models have emerged to counter type 1 diabetes [279]. A robust experimental platform for precise gene editing is provided by isogenic cellular systems and organoids that can be used to develop cultured disease models to investigate complex illnesses like type 1 diabetes and cardiac and cancer-related issues. Using a precise medicine approach, this methodology offers the chance to comprehend the biological pathways and mechanisms in organisms to develop advanced therapies. Models based on isogenic cardiac organoids made from hiPSCs provide a reliable method for examining the cardiotoxicity caused by DOX. A study by Guerrero et al., 2023, highlights using hiPSC-CMs and fibroblasts to develop scaffold-free cardiac organoids [280]. Co-cultivation improves both morphological and functional maturity by enhancing morphology, gene expression (RYR2, SERCA, TNNT2, MYH6), and electrophysiological characteristics. These investigations demonstrate the possible use of cardiac organoids for cardiotoxicity research and disease modeling.

### 5.8. Exploring 3D and 4D Bioprinting Applications

CVD treatment is changing because of recent developments in cardiac tissue engineering and 3D bioprinting technologies [281]. Despite the notable success of conventional therapies like heart transplants and bypass surgeries, issues like donor shortages and high costs still exist. By overcoming these constraints, the 3D bioprinting of engineered heart tissues (EHTs) presents encouraging substitutes [282]. The EHTs use cutting-edge bioprinting techniques that create functional heart tissues using various cell resources and bio-inks [282]. Recent developments have made it possible to create 3D- and 4D-printed cardiac tissues that express essential cellular proteins like connexin-43 and α-actinin and resemble in vivo arrangements. These tissues exhibit heart-like electrophysiological and pharmacological responses while maintaining an anisotropic architecture. Moreover, as seen by the varying reactions to non-cardiotoxic medications when exposed to various organ systems, combining heart, liver, and lung cells utilizing organ-on-a-chip technology improves drug toxicity screening [283]. Recently employed bioprinting techniques like extrusion-based bioprinting create 3D tissue structures layer by layer using bio-inks, offering versatility, cost-effectiveness, and precision in handling large constructs like human-sized organs [284,285]. Furthermore, light-assisted bioprinting uses photoreactive bio-inks that are solidified by light, allowing for acceptable resolutions (5–100 µm) and high printing rates to create intricate tissue structures such as cardiac and vascular networks [286,287]. These 3D- and 4D-printed cardiac systems have potential applications in regenerative medicine, particularly for personalized treatments, drug screening, and developing preclinical disease models, which have gained more attention due to the increasing effectiveness of bioprinting in developing drug discovery models with an emphasis on organoids and bioprinting cardiovascular models [288].

### 5.9. Delivery Approaches

Numerous nano drugs with tumor-targeting capabilities have been developed to optimize DOX transport to tumor tissues and minimize its buildup in cardiac tissue [289]. Although the targeted nanodrugs decrease the concentration of DOX in cardiomyocytes, DOX continues accumulating in cardiomyocytes due to the nanodrugs’ diffusion in many tissues. Since it is impossible to prevent DOX from building up in cardiomyocytes, certain studies have demonstrated using nanoparticles carrying cardioprotective agents that precisely target cardiomyocytes and lessen DOX-induced oxidative damage [290]. For instance, combining doxorubicin with sodium ascorbate (AscNa) in a nanomedicine delivery method maximizes the chemodynamic treatment (CDT) at tumor locations while reducing cardiotoxicity. By taking advantage of the differences in Fe^2+^ levels between the heart and tumors, AscNa protects cardiomyocytes and specifically targets oxidative stress in tumors. This approach guarantees efficient medication administration with fewer adverse effects [290]. Furthermore, an in vivo, poly (lactic-co-glycolic acid) (PLGA)-based nano formulation of DOX demonstrated less cardiotoxicity than both liposomal and conventional formulations. In addition to reducing inflammation and metabolic alterations, PLGA-DOX reduced DIC, suggesting it could be a safer method of delivering chemotherapy [291].

### 5.10. Mathematical Modeling and Multimodal Machine Learning

DOX’s cardiotoxic side effects, which are primarily associated with mitochondrial damage, limit its therapeutic effectiveness and usage. Although their proportional contributions are still unknown, potential mechanisms include ETC inhibition, mtDNA damage, and redox cycling resulting in oxidative stress and damage. Oliveira et al. reported a mathematical-model-based simulation of mitochondrial function modified to account for mtDNA damage and repair, which revealed that ETC inhibition promotes acute toxicity at high DOX dosages (>200 μM) [292]. In contrast, the model suggests a rather minor contribution from DOX redox cycling. Direct damage to mtDNA causes persistent and lasting mitochondrial malfunction, the primary source of chronic toxicity at treatment dosages. Another study by Fernandez et al. utilized revised mathematical models to identify the roles that channels, pumps, and transporters play in the cardiotoxicity of DOX, which links the pharmacological effects on specific ion channels to phenotypic outcomes like action potential duration (APD) and intracellular calcium concentration. According to the study, both acute and chronic DOX exposure showed a rise in K^+^ permeability through the rapid delayed resolving K^+^ current (IKr), enhanced sarcoplasmic reticulum (SR) Ca^2+^ leakage, and did not impair Na^+^/Ca^2+^ exchangers during DOX exposure [293]. As the knowledge progresses and the mathematical models improve, such studies may significantly increase our understanding of the mechanisms of DOX-induced cardiotoxicity at the molecular level and fill in a significant gap addressing the role of the interactions between altered ion channels during cardiotoxicity.

### 5.11. Designing Robust Clinical Trials

Several investigations emphasize how sex affects CTRTOX. For example, the results of many preclinical studies in rats indicated that males are more susceptible to DIC than females, and differences in mitochondrial dysfunction and altered energy signaling pathways primarily drive this sexual dimorphism. In males, doxorubicin treatment leads to severe mitochondrial impairment, downregulation of mitochondrial biogenesis, and altered cardiolipin homeostasis. In contrast, females exhibit better mitochondrial preservation and AMPK activity, which may protect them from cardiotoxic effects. These findings suggest that targeting mitochondrial function and energy metabolism pathways could help mitigate doxorubicin-induced heart damage, with a potential focus on sex-specific therapeutic strategies [294]. In a mice model of DIC, TRPC6 deficiency provides cardioprotection against doxorubicin-induced cardiac damage and cardiomyopathy in male mice, highlighting that TRPC6 plays a critical role in promoting cardiac toxicity during chemotherapy. The study also reveals significant sex differences, with female mice being less susceptible to doxorubicin-induced heart damage, possibly due to the protective effects of estrogen on TRPC channel activity [295]. Women comprised 75.5% of the patients in a meta-analysis looking at proper ventricular alterations in CTRTOX [296]. This highlights the significance of taking sex into account as a variable in CTRTOX investigations, although it does not directly prove a causal relationship between sex and higher risk. Additionally, a study of the differences between genders in cardio-oncology points out that little is known about how sex affects cardiotoxicities, indicating a substantial research gap [297]. The authors stress that sex probably affects how the body reacts to cardiovascular and oncologic treatments and the fundamental bases of cancer susceptibility [297]. To completely comprehend the cardiovascular effects of oncological therapy, more gender-specific research is required due to this intricate interaction. It is also essential to investigate how specific cancer treatments interact with sex. Regardless of sex, a study assessing the risk of cardiotoxicity linked to associated trastuzumab emtansine (TDM1) and radiotherapy in individuals with early-stage HER2-positive breast cancer observed no apparent change in global longitudinal strain (GLS) or LVEF following radiation [298]. Nevertheless, the limited sample size restricts how broadly these results can be applied. Independent of sex, Asian race was a significant predictor of CTRTOX, especially by GLS criteria, according to another study that looked at racial and ethnic differences in anthracycline and HER2-targeted cardiotoxicity in patients with breast cancer [299]. Because treatment groups differ in how they order echocardiograms, the authors admit confounding factors in detection. A post hoc evaluation of the GO2 trial examined the association between survival and estrogen receptor levels in advanced gastric adenocarcinoma in the setting of hormone treatments [300]. Female sex was linked to better overall survival, even though estrogen receptor expression did not significantly affect survival [300]. This demonstrates the possibility of sex-related variations in survival and the response to treatment that are not influenced by hormone receptor status. More investigation is required to identify the underlying mechanisms causing these sex- and race-based disparities.

## 6. Conclusions

As we elucidate the potential of 2D and 3D CM models from hiPSCs for the predictive modeling of cardiotoxicity to study the molecular mechanisms in response to CTRTOX and screen for potential new therapeutic interventions, we continue to expand our knowledge. This review provides a detailed discussion of the pathophysiology and current understanding of the mechanisms associated with CTRTOX and the recent advances in the therapeutic strategies in different animal models and clinical trials to prevent CTRTOX. In conclusion, the path ahead for hiPSC-CM research and its extensive application in understanding the mechanisms of CTRTOX is promising. Identifying epigenetic markers, integrating novel pharmacological compounds and therapeutic strategies, and well-designed clinical studies will pave the way for precision medicine.


## Figures and Tables

**Figure 1 ijms-26-03966-f001:**
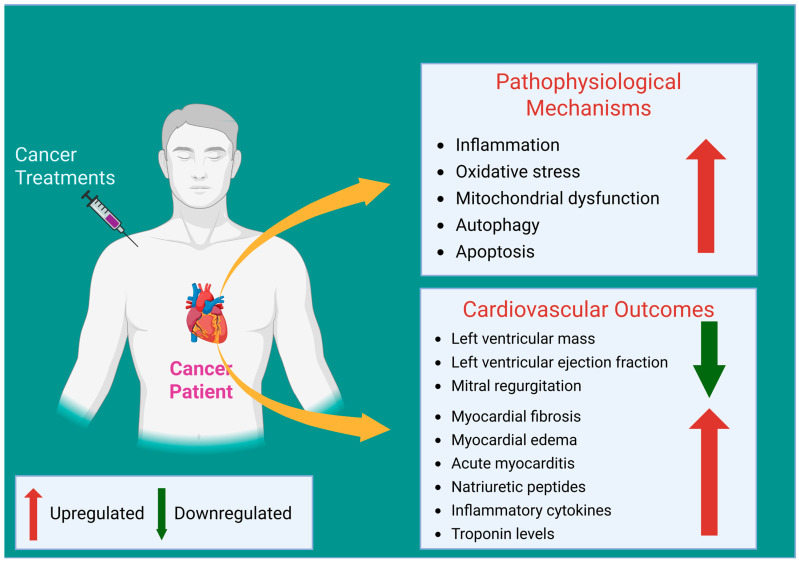
Schematic diagram illustrating cardiovascular complications during and after cancer therapy.

**Figure 2 ijms-26-03966-f002:**
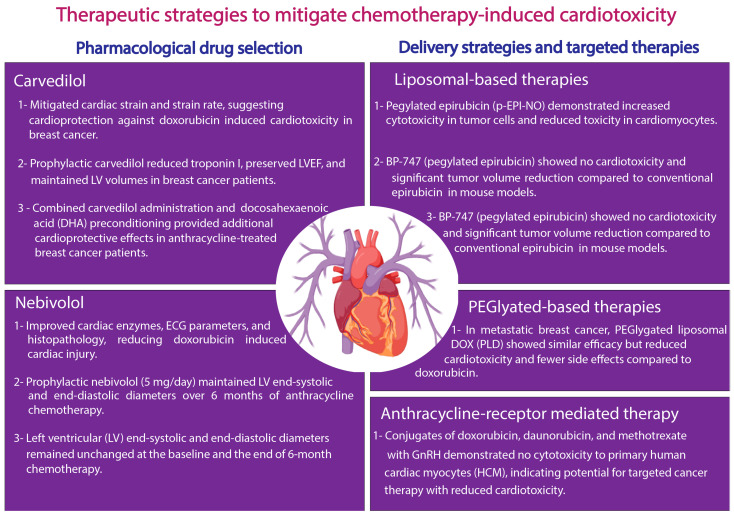
Schematic depiction of the current therapeutic strategies to mitigate cancer-treatment-related cardiotoxicity.

**Figure 3 ijms-26-03966-f003:**
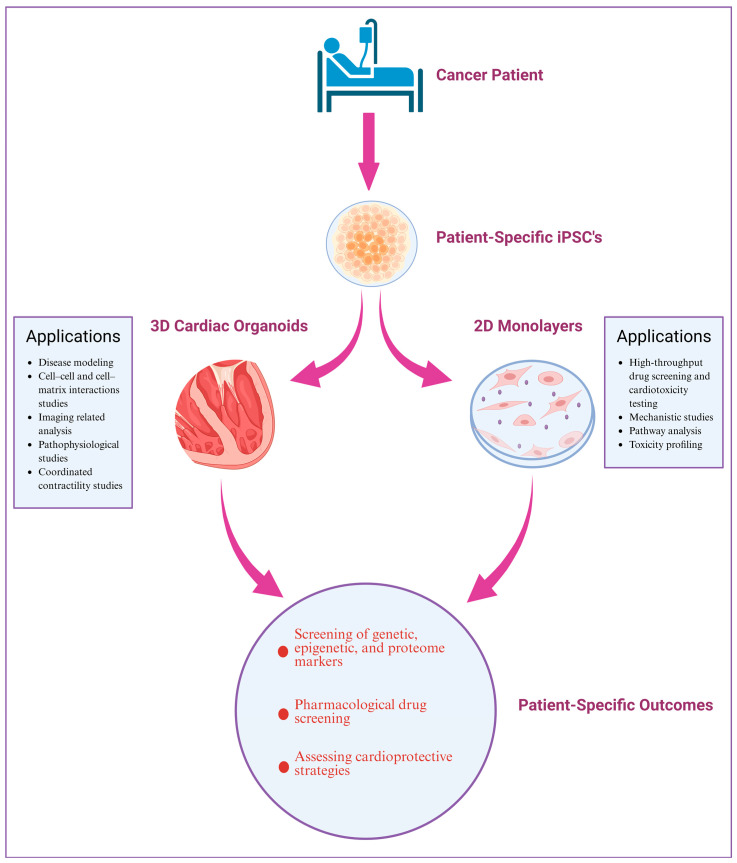
Human iPSC-CMs and their application in modeling cancer-treatment-related cardiotoxicity.

**Figure 4 ijms-26-03966-f004:**
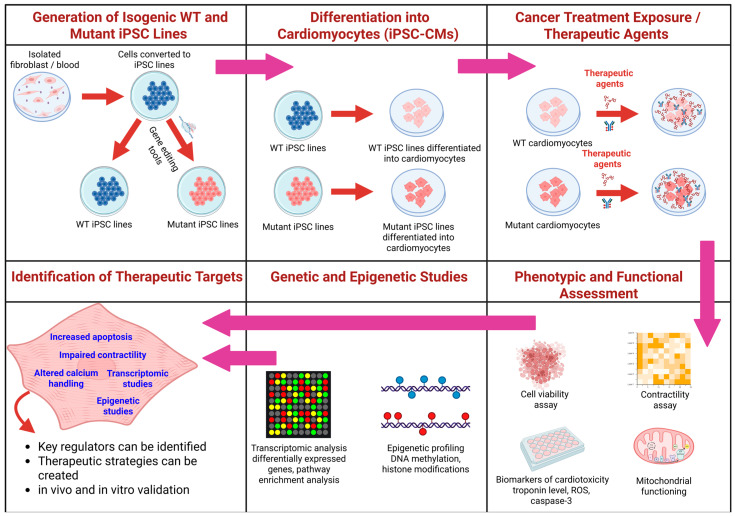
Schematic diagram illustrating the application of hiPSC-CMs from healthy and cancer patients for the identification of genetic, epigenetic markers and therapeutic targets.

**Table 1 ijms-26-03966-t001:** Evidence of an effect of chemotherapy-induced cardiotoxicity.

Chemotherapy Drugs	Experimental Model	Dosage	Effect/Mechanisms	References
Doxorubicin (DOX)	Mouse leukemia L1210 cells	0–5 µM (1 h incubation at 37 °C)	Iminodaunorubicin produces transient DNA breaks; Adriamycin induces persistent DNA breaks	[41]
	Rat cardiomyocytes	40–160 µM (20 min incubation at 37 °C)	Increased oxidative stress generation near mitochondria to induce cardiotoxicity	[42]
	Bovine heart submitochondrial preparations	25–50 µM (15 min incubation at 37 °C) and 0.5–1.0 mM (2 h incubation at 37 °C)	Low concentrations induce oxidative damage via redox cycling, inactivating NADH oxidase	[43]
			Higher concentrations cause non-oxidative inactivation of ETC complexes by cardiolipin binding	
	Rat cardiomyocytes	0.1–2 µM (24–48 h incubation at 37 °C)	Impairment of cellular energetics and reduction in ATP levels, affecting mitochondrial respiration and cell viability	[44]
	Bovine aortic endothelial cells	0.5–2 µM (12–24 h incubation at 37 °C)	Increased eNOS transcription and oxidative-stress-mediated apoptosis	[45]
	Rat cerebellum homogenates	Doxorubicin (Ki = 24 µM), Aclarubicin (Ki = 50 µM)	Non-competitively inhibits nitric oxide synthase (NOS), causing cardiovascular toxicity	[46]
	Purified endothelial nitric oxide synthase	5–100 µM	Induces eNOS-dependent superoxide generation via doxorubicin redox cycling, causing cardiotoxicity	[47]
	Rabbit aortic ring segments and human brachial artery	10 mg/kg; intravenous single dose	Rapid attenuation of endothelial-dependent dilation, increased superoxide generation via eNOS dysfunction, and rapid depletion of systemic NO levels lead to endothelial dysfunction	[48]
	Purified nitric oxide synthase and rat cerebellar homogenates	50 µM Adriamycin	Generation of superoxide radicals via (NOS), causing dysfunction in NOS activity	[49]
	(Nox2−/−) and wild-type (WT) littermate mice	12 mg/kg DOX or saline control by three weekly intraperitoneal injections (4 mg/kg at 0, 7, and 14 days)	Induced cardiomyopathy characterized by oxidative stress, cardiomyocyte apoptosis, myocardial dysfunction, interstitial fibrosis, and cardiac remodeling via increased Nox2-derived ROS and upregulated mitofusin-2 expression	[50]
	C57BL/6 mice	8 mg/kg weekly; intraperitoneal injection, once a week, for 4 weeks; (total 32 mg/kg cumulative dose)	Cardiac fibrosis, apoptosis, cardiomyocyte injury, oxidative stress induction through upregulation of NOX2 and NOX4. Protective effects observed by Astragaloside IV treatment reducing oxidative stress	[51]
	Langendorff-perfused Wistar rat hearts and hearts from doxorubicin-treated Wistar rats	5–25 µM (during 80 min perfusion, ex vivo) and intraperitoneal administration of 2 mg/kg doxorubicin or an equivalent volume of saline was administered to animals via the implanted catheter three times a week for 2 weeks for a total dose of 12 mg/kg	Inhibition of AMPK despite energetic stress and activation of Akt and MAPK via DNA damage signaling resulted in increased mTOR activation, contributing to chronic cardiac dysfunction and remodeling	[52]
Daunorubicin (DNR)	Acute myeloid leukemia (AML) patients	50–630 mg/m^2^ cumulative dose	Identification of POR gene variants as potential markers of DNR-induced cardiotoxicity; the genetic factors account for a significant proportion of LVEF drop via dose–genotype interaction	[53]
	Eight-week-old male Sprague-Dawley rats	Cumulative dose of 9 mg/kg administered at 3 mg/kg intravenously, given in three equal injections at 48-h intervals	Induces cardiotoxicity and nephrotoxicity manifested as interstitial edema, subendocardial fibrosis, perinuclear vacuolation, and myocardial degeneration, and carvedilol co-administration provides protection	[54]
	Ten–twelve-week-old male Wistar rats (comparative acute vs. subchronic cardiomyopathy models)	Acute—Six intraperitoneal injections of 3 mg/kg every 48 h; Subchronic—Single intravenous injection of 15 mg/kg	Both models show reduced left ventricular weight and function with upregulation of natriuretic peptides. The subchronic model shows decreased Myh6 expression and altered stem cell marker expression indicative of impaired regenerative potential	[55]
	Ten–twelve-week-old male Wistar rats	Six intraperitoneal doses of 3 mg/kg every 48 h	Early cardiomyopathy characterized by depressed left ventricular pressure and contractility, a twofold upregulation of ryanodine receptor 2 (RyR2), increased NPPA/NPPB expression, and decreased alpha-tubulin gene expression suggesting the disruption of Ca^2+^ handling in cardiomyocytes	[56]
	Rat H9c2 cardiomyocytes (in vitro study)	Cells pre-treated with Puerarin (1–100 µg/mL) for 24 h, then exposed to 1 µM DNR for an additional 24 h	DNR induces apoptosis by increasing Ca^2+^⁺ influx and activating caspase-3 via the PI3K/Akt signaling pathway. Puerarin pre-treatment attenuates these effects by inhibiting Ca^2+^⁺ influx and enhancing p-Akt activation, thereby protecting cells from apoptosis	[57]
Epirubicin (EPI)	Male Wistar rats	Single dose of 10 mg/kg; intraperitoneal administration	Induces mitochondrial degeneration, swelling, intracytoplasmic vacuolization, and focal myofilament disarray in cardiomyocytes, leading to cardiotoxicity	[58]
	Male Sprague-Dawley rats	8 mg/kg; intraperitoneal administration	Induces cardiotoxicity via the upregulation of genes promoting autophagy and apoptosis	[59]
	Non-Hodgkin lymphoma patients	Bolus dose of 40 mg/m^2^, intravenous infusion (administered for 3 weeks)	Increases QT dispersion and QTc dispersion, reflecting early electrophysiological alterations linked to cardiotoxicity	[60]
	Cancer patients in a phase II trial	Cumulative dose of 200 mg/m^2^	Induces impairment in systolic left ventricular function, characterized by a reduction in strain rate peak at 3-, 6-, 12-, and 18-months follow-up and correlates with increased IL-6 and oxidative stress markers	[61]
	Breast cancer patients	Combination of epirubicin: 100 mg/m^2^ and cyclophosphamide: 600 mg/m^2^ at about 3-week intervals for eight cycles	Upregulates apoptosis and downregulates 5′-aminolevulinate synthase 2 (ALAS2), implicating glycine, serine, and threonine metabolism disruptions in cardiomyopathy development	[62]
Idarubicin	Acute myeloid leukemia (AML) patient without cardiac risk factors	12 mg/m^2^/day for 3 days (cumulative dose: 36 mg/m^2^) administered via 30 min intravenous infusion	First exposure resulted in severe subacute congestive heart failure (CHF) and ventricular tachycardia (VT), a prolonged QTc interval with frequent premature ventricular contractions (QTc ~400 ms)	[63]
	Acute myeloid leukemia (AML) patient	12 mg/m^2^/day for 3 days (cumulative dose: 36 mg/m^2^) administered via 30-min intravenous infusion	Induced cardiac arrest shortly after the first dose of idarubicin in an AML patient	[64]
	AML patient without cardiac risk factors	Cumulative dose of 36 mg/m^2^ administered within 2 weeks	Induced cardiomyopathy with a 25% reduction in left ventricular ejection fraction (LVEF), impaired right ventricular (RV) function, and severe mitral regurgitation	[65]
Mitoxantrone (MTX)	Stage IV breast cancer patients	Mitoxantrone 10 mg/m^2^, methotrexate 40 mg/m^2^, 5-fluorouracil 600 mg/m^2^, given every 3 weeks	Demonstrated broad antitumor activity with a favorable toxicity profile (notably lower cardiotoxicity) compared to other anthracyclines	[66]
	In vivo in MTX-treated rats and in vitro in H9c2 cells	Not Specified	MTX and its naphthoquinoxaline metabolites accumulate in the heart and liver. CYP450- and CYP2E1-mediated metabolism contributes to cytotoxicity in H9c2 cells	[67]
	7-day differentiated H9c2 cells (in vitro)	0.01–5 µM	MTX alters energetic pathways (e.g., increased ATP and decreased lactate production), while its NAPHT metabolite is less cardiotoxic, indicating differences in metabolic disruption	[68]
	HL-1 cardiomyocytes (in vitro)	0.1, 1, and 10 µM MTX	Impairs proteasome activity and triggers early energetic and proteomic changes that disturb oxidative stress homeostasis	[69]
	Adult and infant mice (in vivo)	Adult mice—7.0 mg/kg cumulative dose; Infant mice—7.0 mg/kg cumulative dose (protocols adjusted for age)	In adult mice, MTX induced myocardial injury, fibrosis, and the increased expression of NF-κB p52 and TNF-α, and decreased IL-6 expression, implicating inflammation in cardiotoxicity; infants showed higher resilience	[70]
	Adult CD-1 male mice (in vivo)	6 mg/kg cumulative dose	Decreases in AMPK and GAPDH content; decreased free carnitine (C_0_) and increased acetyl carnitine (C_2_) suggest a shift toward fatty acid oxidation	[71]
	AML patient (case report)	Not Specified	Induced acute myocarditis characterized by a 25% reduction in LVEF, diffuse myocardial edema, and delayed gadolinium enhancement, demonstrating acute cardiotoxicity	[72]

**Table 2 ijms-26-03966-t002:** Current therapeutic strategies to mitigate chemotherapy-induced cardiotoxicity.

Therapeutic Strategies	Experimental Model	Dosage	Cardiovascular Outcome	References
Carvedilol Prophylaxis	Breast cancer patients receiving anthracycline-based chemotherapy	6.25 mg carvedilol daily (during chemotherapy)	Mitigation of cardiac strain and strain-rate changes and prevention of DOX-induced cardiotoxicity	[199]
	Recently diagnosed breast cancer patients	Prophylactic carvedilol (specific dosage not mentioned)	Reduced troponin I level, preserved LVEF, and favorable changes in LVES and LA diameter and the inhibition of AIC	[200]
	HER2-negative breast cancer patients with normal LVEF undergoing anthracycline treatment	6.25 mg carvedilol daily	Significant reduction in troponin levels and diastolic dysfunction	[201]
	Breast cancer patients receiving anthracycline treatment	Carvedilol + DHA (administered starting 1 week before and continued for 90 days)	Attenuation of subclinical cardiotoxicity, evidenced by smaller declines in LVEF (measured by CMR), preservation of global longitudinal strain (via ECHO), lower elevations in hs-cTnT and NT-proBNP, and reduced QTc prolongation compared to the placebo	[202]
Nebivolol Prophylaxis	Preclinical study in rats	1–2 mg/kg orally in rats	Improvement in heart index, cardiac enzymes, histopathology, and ECG parameters, reduction in doxorubicin-induced cardiotoxicity	[203]
	Breast cancer patients	5 mg nebivolol daily	Protection of myocardium with stable LV end-systolic and end-diastolic diameters over 6-month chemotherapy	[204]
Dexrazoxane Prophylaxis	Preclinical study in rats	62.5 mg/kg intraperitoneally at 0, 3, and 6 h	preservation of left ventricular function, lower rates of cardiomyopathy, and low oxidative stress markers compared to untreated groups	[205]
	Systematic review containing 23 clinical trials, which comprised a total of 14,652 subjects	Not Specified	Reduced cardiovascular risk, 50% risk reduction, threefold lower chance of cardiac dysfunction, 74% decrease in cardiotoxicity	[206]
Mitochondrial-Targeted Antioxidants (MTAs)	Spontaneously hypertensive rat (SHR) implanted with the SST-2 breast tumor cell line	Doxorubicin (DOX), 4 mg/kg, administered intraperitoneally (IP) once weekly for 3 weeks; Mito-Tempol (Mito-T), 1 mg/kg/day, administered intraperitoneally (IP) daily Dexrazoxane (DXZ), 100 mg/kg, administered intraperitoneally (IP) 30 min before DOX injection	Mito-Tempol and dexrazoxane protected the heart from DOX-induced toxicity, preserving LVEF and reducing oxidative stress and apoptosis	[207]
	AGS gastric cancer cell line (in vitro) and male BALB/c nude mice xenografted with AGS gastric cancer cells (in vivo)	Mito-FF peptide, 10 mg/kg, administered intravenously (IV) every 3 days; 5-Fluorouracil (5-FU), 30 mg/kg, administered intraperitoneally (IP) every 3 days	Mito-FF peptide protected against 5-FU-induced cardiotoxicity, reducing ROS generation and apoptosis in cardiac tissues	[208]
PEGylated Liposomal Doxorubicin (CAELYX)	Preclinical pharmacokinetic studies in animals (rats and beagle dogs)	0.5 mg/kg	Reduced myocardial accumulation, suggesting a reduced risk of cardiotoxicity, apoptosis, and oxidative stress in the heart tissue	[209]
	In vivo mouse model (BALB/c mice with implanted tumors)	Single IV bolus of 10 mg/kg for free and liposomal DOX formulations	Approximately fourfold lower doxorubicin concentrations in heart tissue compared to tumor tissue; liposomal formulation effectively minimizes cardiac drug deposition and, consequently, DOX-induced cardiac injury	[210]
	Pilot clinical study in cancer patients (*n* = 16, various tumor types)	Free DOX and liposomal DOX (Doxil) at dose levels of 25 mg/m^2^ and 50 mg/m^2^	Significantly reduced cardiac accumulation of doxorubicin, preserved left ventricular ejection fraction (LVEF); minimizes myocardial exposure, thereby protecting against DOX-induced cardiotoxicity	[211]
PEGylated Liposomal Doxorubicin (PLD)	Phase III clinical trial in women with metastatic breast cancer	PLD—50 mg/m^2^ every 4 weeks; Conventional DOX—60 mg/m^2^ every 3 weeks	Preserved left ventricular ejection fraction (LVEF) with a threefold lower incidence of cardiotoxic events compared to conventional DOX	[212]
	Retrospective study in HER2-positive early breast cancer patients.	50 mg/m^2^ every 4 weeks	Clinical cardiotoxicity occurred in 8.6% of patients, subclinical events in 24.3% (10–16% decline), with no treatment interruptions	[213]
Liposomal Daunorubicin Delivery	Pediatric acute myeloid leukemia induction trial	L-DNR, 80 mg/m^2^ per day for 3 days; Idarubicin, 12 mg/m^2^ per day for 3 days	Only one patient in the L-DNR arm and three in the idarubicin arm developed subclinical or mild cardiomyopathy	[214]
PEGylated Epirubicin with NO-Releasing Moiety (p-EPI-NO)	In vivo mouse models bearing Caco-2 and SKOV-2 tumors	Not Specified	95% reduction in tumor volume while virtually eliminating clinical, anatomical, and biochemical signs of cardiotoxicity compared to free epirubicin and conventional PEGylated epirubicin	[215]
Mitoxantrone (MTO) formulated in Cardiolipin-based Anionic Liposomes	Preclinical intraperitoneal chemotherapy model	Not Specified	Prolonged retention of MTO at tumor sites, resulting in negligible cardiotoxicity versus free MTO	[216]
Carvedilol (CAR), Resveratrol (RES), and Liposomal Resveratrol (LIPO-RES)	Rat model of DOX-induced cardiomyopathy	DOX—2 mg/kg twice per week (weeks 2–6);	The combination, particularly CAR/LIPO-RES, significantly reduced serum CK-MB, troponin-I, and LDH levels, preserved cardiac histology, increased S100A1 and SERCA2a expression, and mitigated oxidative stress and inflammation	[217]
		CAR—30 mg/kg; RES and LIPO-RES: 20 mg/kg for 6 weeks		
Liposomal Resveratrol (LIPO-RES)	In vivo study in rats	20 mg/kg for 6 weeks	Protection against DOX-induced oxidative stress, inflammation, and calcium dysregulation	[217]
Idarubicin-Loaded Solid Lipid Nanoparticles	In vivo study in rats	2 mg/kg	Reduced drug uptake in the heart, lowering cardiotoxic potential	[218]
GnRH-based Conjugates Containing Doxorubicin, Daunorubicin, and Methotrexate	Human cardiac myocytes (HCMs) and human umbilical vein endothelial cells (HUVECs)	Various concentrations tested	No cytotoxic effect on cardiomyocytes	[219]

**Table 3 ijms-26-03966-t003:** Modeling chemotherapy-induced cardiotoxicity using hiPSC-CMs.

Chemotherapy Drugs	Experimental Procedure	Major Findings	Limitations	References
Doxorubicin (DOX)	hiPSC-CMs derived from breast cancer patients (with/without DIC) were exposed in vitro to doxorubicin (0.1–10 µM) for 24–72 h. The assessments included cell viability, mitochondrial/metabolic function, calcium handling, and transcriptomic profiling	hiPSC-CMs from patients who developed doxorubicin-induced cardiotoxicity (DIC) showed markedly lower viability, impaired mitochondrial function, disrupted calcium handling, decreased antioxidant activity, and increased ROS production compared to cells from non-DIC patients	Limited number of patient-specific lines and potential variability in differentiation efficiency may affect reproducibility	[229]
	hiPSC-CMs were evaluated using multielectrode array (MEA) analyses for electrical activity (beating rate, field potential duration) and high-content imaging for mitochondrial parameters under single (3 h, 2 days) and repetitive dosing (3 cycles of 2 days each, with washout up to day 14)	Acute exposure up to 6 µM did not alter electrical activity, whereas chronic exposure at nanomolar concentrations significantly reduced cell viability and induced subtle changes in mitochondrial membrane potential and calcium levels	In vitro endpoints might not capture all aspects of long-term cardiotoxicity	[230]
	Repeated exposure of hiPSC-CMs to DOX combined with NMR-based metabolic profiling to assess changes in cellular metabolism	Repeated doxorubicin exposure resulted in decreased utilization of pyruvate and acetate, with a concomitant accumulation of formate	NMR sensitivity limits the detection of low-abundance metabolites; further validation	[231]
	Low dose of doxorubicin (100 nM for 14 days) in vitro, while in parallel, a rat model was used for in vivo studies	26% reduction in hiPSC-CM viability	The chronic in vitro exposure model may not fully recapitulate the long-term, multifactorial nature of cardiotoxicity in patients; translation to clinical outcomes remains uncertain	[232]
Tyrosine Kinase Inhibitors (TKIs)	hiPSC-CMs, hiPSC-derived endothelial cells, and cardiac fibroblasts. The assessment includes cell viability, contractility, electrophysiology, and the generation of a “cardiac safety index”	VEGFR2/PDGFR-inhibiting TKIs (sorafenib, regorafenib, ponatinib) exhibited the highest cardiotoxicity (LD_50_ around 3 µM), and co-treatment with exogenous insulin or IGF1 improved cardiomyocyte viability	In vitro screening may not capture complex in vivo pharmacodynamics and inter-patient variability	[233]
Trastuzumab	hiPSC-CMs were treated with 10 μM doxorubicin, 1 μM trastuzumab, 1 ng/mL NRG-1, and 100 ng/mL HB-EGF in a monoculture and in co-culture with endothelial cells	TZM disturbed the calcium balance without causing any cell death; adding neuregulin-1 or HB-EGF restored the normal functioning of cells	In vitro findings require validation in clinical settings; the complexity of energy metabolism in vivo may not be fully replicated in hiPSC-CMs	[234]
	hiPSC-CMs derived from both healthy individuals and patients with breast cancer who experienced trastuzumab-induced cardiac dysfunction were chronically exposed to trastuzumab. Functional assays include contractility, calcium handling, RNA-seq transcriptomics, and metabolic modulation experiments	Clinically relevant doses of trastuzumab impaired contractility and calcium handling without causing cell death. The transcriptomic analysis revealed mitochondrial dysfunction and altered energy metabolism. hiPSC-CMs from patients with severe cardiotoxicity were more vulnerable, and metabolic modulators improved the phenotype—suggesting that targeting altered energy metabolism may be therapeutic	Small sample size of patient-specific lines and inherent variability in hiPSC-CM maturation	[235]

**Table 4 ijms-26-03966-t004:** Screening and personalized medicine using hiPSC-CMs.

Chemotherapy Drugs	Experimental Procedure and Dosage	Methodology	Genetic/Epigenetic Markers	Major Findings	References
Doxorubicin (DOX)	156 nM for a 2-day exposure period (DOX-Day2) or three consecutive exposure periods of DOX for 6 days (DOX-Day6)	miRNA microarray	miR-187-3p, miR-182-5p, miR-486-3p, miR-486-5p, miR-34a-3p, miR-4423-3p, miR-34c-3p, miR-34c-5p, miR-1303, miR-182-5p, miR-4423-3p, and miR-34c-5p	(1) Upregulation of miR-34c-3p and miR-34c-5p may be early indicators of cardiac dysfunction, the development of cardiac pathologies, and future heart failure; (2) miR-187-3p overexpression in cardiomyocytes might be the result of DOX-induced DNA damage; (3) DOX-induced early upregulation of miR-486-5p suggests that the elevation of miR-486-5p may be considered as an early event in the development of cardiovascular diseases	[237]
	Day 30 hiPSC-CMs were treated for 24 h or 72 h with doxorubicin (0.01–100 μM)	RNA-seq gene expression analysis; Quantitative real-time PCR	SNP (rs2229774) in retinoic acid receptor-γ (RARG)	(1) A rs2229774 patient-specific model recapitulates the patients’ cardiotoxicity phenotype; (2) rs2229774 increased doxorubicin-induced cardiotoxicity susceptibility; (3) RARG agonists attenuate cardiotoxicity without affecting chemotherapy efficacy; (4) Pharmacogenetic screening for rs2229774 and RARG agonists is a clinical option	[239]
	Patient-specific DIC was characterized	SLC28A3 locus genetic fine mapping using a MinION nanopore sequencer and RNA sequencing	SNP rs11140490 in the SLC28A3 locus	rs11140490 in the SLC28A3 locus was linked to reduced DIC	[240]
	Nine hiPSC-CM lines intrinsically polymorphic (3 lines/genotype) at rs28714259 were treated with the vehicle or 100 nM dexamethasone for 24 h followed by the vehicle or 1 µM doxorubicin treatment for 24 h	RNA sequencing gene expression	SNP rs28714259 in the GR:rs28714259 locus	rs28714259 was linked to anthracycline-induced cardiotoxicity through disruption of GR:rs28714259 locus binding and its subsequent protective signaling	[242]
	hiPSC line 19c3 was generated from peripheral blood mononuclear cells from a healthy individual. Effect of doxorubicin (72 h) on hiPSC-CM viability in the control (isotype) and knockouts for EXOC6B, FCHSD2, NIPAL2, SYNPO2, PDXK, SLMAP, and RORA was assessed. Day 30 hiPSC-CMs were treated for 72 h with doxorubicin (0.01–100 μM)	Epigenome-wide association study (EWAS); DNA-methylation EPIC array	EXOC6B, FCHSD2, NIPAL2, SYNPO2, PDXK, and SLMAP	Knockout of genes EXO6CB, FCHSD2, NIPAL2, and SYNPO2 in hiPSC-CMs increased sensitivity to doxorubicin	[243]

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
