# Peer review of "Understanding the Mechanisms of Chemotherapy-Related Cardiotoxicity Employing hiPSC-Derived Cardiomyocyte Models for Drug Screening and the Identification of Genetic and Epigenetic Variants"

_ijms, 2025, doi:10.3390/ijms26093966_

Round 1

Reviewer 1 Report

Comments and Suggestions for Authors

In this manuscript, Solomon and colleagues address the mechanisms of chemotherapy-related cardiotoxicity and the potential of using human induced pluripotent stem cell-derived cardiomyocyte models for drug screening and developing personalized cardioprotective strategies.

This is a thorough and well written document and a very timely-addressed topic. The several sections are well distributed and complementary contribute to an up-to-date information and discussion on this topic. The authors describe the mechanisms of chemotherapy-related cardiotoxicity (CTRTOX), discussing the role of several chemotherapy drugs, particularly anthracyclines and others, that can cause severe cardiotoxicity through mechanisms such as oxidative stress, DNA damage, mitochondrial dysfunction, and apoptosis. ​They nicely discuss several of the cardiovascular complications during and after cancer therapy.

Then, they extensively present and discuss the use of human iPS Cells for modeling cardiac diseases. One of the main conclusions and message of this manuscript is that despite the promise of hiPSCs-CMs in studying CTRTOX, challenges remain, including the need for scalable manufacturing processes, improved models that incorporate mechanical stress, and robust clinical trials to validate findings.

Recommendations:

  1. Tables 1 and 2 although very informative are of difficult access to the reader. Lettering should be in a higher font size.
  2. Figure 3, This figure is expected to illustrate the use of human induced pluripotent stem cell-derived iPSC-derived cardiomyocyte models in studying CTRTOX. The authors should include here examples to both 2D and 3D models, as they are so extensively eluded in the text. Moreover, current researchdemonstrates the benefits towards the use of 3D cardiac microtissues or of cardiac organoids for cardiotoxicity research and disease modelling.

Author Response

Itemized responses to the reviewers:

All changes and additions in the revised review manuscript are indicated in red.

# Reviewer 1

In this manuscript, Solomon and colleagues address the mechanisms of chemotherapy-related cardiotoxicity and the potential of using human induced pluripotent stem cell-derived cardiomyocyte models for drug screening and developing personalized cardioprotective strategies.

This is a thorough and well written document and a very timely-addressed topic. The several sections are well distributed and complementary contribute to an up-to-date information and discussion on this topic. The authors describe the mechanisms of chemotherapy-related cardiotoxicity (CTRTOX), discussing the role of several chemotherapy drugs, particularly anthracyclines and others, that can cause severe cardiotoxicity through mechanisms such as oxidative stress, DNA damage, mitochondrial dysfunction, and apoptosis. ​They nicely discuss several of the cardiovascular complications during and after cancer therapy.

Then, they extensively present and discuss the use of human iPS Cells for modeling cardiac diseases. One of the main conclusions and message of this manuscript is that despite the promise of hiPSCs-CMs in studying CTRTOX, challenges remain, including the need for scalable manufacturing processes, improved models that incorporate mechanical stress, and robust clinical trials to validate findings.

Recommendations:

Comment 1: Tables 1 and 2 although very informative are of difficult access to the reader. Lettering should be in a higher font size.

Response 1: Thank you for your valuable comment, we have revised all the tables to increase the font sizes, resolution to enhance the readability. Please refer to the revised manuscript.

Comment 2: Figure 3, This figure is expected to illustrate the use of human induced pluripotent stem cell-derived iPSC-derived cardiomyocyte models in studying CTRTOX. The authors should include here examples to both 2D and 3D models, as they are so extensively eluded in the text. Moreover, current research demonstrates the benefits towards the use of 3D cardiac microtissues or of cardiac organoids for cardiotoxicity research and disease modelling.

Response 2: We appreciate the reviewer’s comment. As suggested, we have revised Figure 3 to demonstrate the applications of both 2D and 3D models. Please refer to the revised Figure 3 in the manuscript.

Reviewer 2 Report

Comments and Suggestions for Authors

The main objective of the review is to discuss the use of hiPSC-derived cardiomyocyte models for drug screening and the identification of genetic and epigenetic variants, which begins in topic 4. However, until then, the authors provide an extensive and detailed review of other basic processes such as "CTRTOX Pathophysiology and Mechanisms" and "Recent Advancement in Therapeutic Strategies in the Treatment of CTRTOX", accompanied by tables and figures.

The main content of the manuscript is interesting and well detailed, but the other sections with excessive content impair the reader's experience. In addition, the authors need to review all the tables (sent in image form): the sentences are unformatted with centered texts at some points (such as in the therapeutic strategies column in Table 2). Sentences with periods appear at various times.

Several issues need to be reviewed:

  1. The title needs modifications, in the current version it has ; and an acronym (hiPSC);
  2.  Lines 46-48: "Furthermore, recent negative clinical trials have stymied prophylactic strategies to avert CTRTOX, whose devastating cardiovascular risks persist for months to years after completion of chemotherapy." This sentence needs a reference to clinical trials.
  3. Line 98: "demonstrated that (iPSCs) can be derived". It is necessary to remove the parentheses.
  4. Line 124: "human induced pluripotent stem cell-derived CMs (h-iPSC-CMs)". Throughout the manuscript, the authors do not use the hyphen. Furthermore, to talk about iPSC-CMs in other organisms (in line 106) the acronym would need to be defined.
  5. "cancer treatment related cardiotoxicity" can be used in the figure legend, but does not need to be in the image.
  6. What does S. No. mean in Table 1? Why does DOX appear with four numbers in table 3?
  7. Line 172, 175: "hydrogen peroxide (H2O2) " "H2O2 production [61]. H2O2". The number 2 needs to be subscripted.
  8. The authors use et al. several times in the manuscript, but 3 times they use "and colleagues". Line 245: " Kucerova and colleagues"; Line 574: " Imbaby and colleagues"; Line 782: " Tomoya and colleagues". This needs to be standardized.
  9. Several subitems have : as in "2.1.4. Idarubicin:", "2.2.1. Trastuzumab:", and "3.2. Delivery Strategies and Targeted Therapies:". This needs to be revised.
  10. In section "2.2." the authors put the drug name with abbreviation in the title "2.2.2. Lapatinib (LAP):" while in other compounds not "2.2.3. Sunitinib". It is necessary to standardize.
  11. It would be interesting to summarize the information in "Table 1. Evidence of an effect of chemotherapy-induced cardiotoxicity." Of the 33 studies, only 7 (21%) involved patients, 11 involved cells (33%), which is precisely the focus of the review (using cell models for drug screening and the identification of genetic and epigenetic variants).
  12. In Figure 2, the authors use the citation number in the text. Since the image should be representative of the manuscript text, the authors should remove the references or cite them by name. This is different from a citation in a table, for example. In addition, numbers could be replaced by topic symbols.
  13. The tables are as images, so the resolution is not satisfactory.
  14. In some sentences of Table 3, periods appear at the end. Authors should standardize this, including justified alignment. The table seems very disfigured and without resolution. Again, another table presented in the form of a figure. The text on the last line is in a different position than the others. In addition, two sentences have periods. The table needs to be reviewed.
  15. The authors use "1. Introduction", but in the following sections "2.0. CTRTOX Pathophysiology and Mechanisms" and "4.0. Application of Human iPSCs-Derived Cardiomyocytes to Study CTRTOX" for example. The topic numbers should be 2 and 4, 2.1 is a subtopic, and there is no subtopic 2.0.
  16. Topic 6 "Conclusion and Perspectives" needs to be completely reformulated. This concluding topic needs to be short and should address the main conclusions and perspectives in the field. The authors address new information. If they deem it necessary, this information should appear in previous topics.
  17. The "Abbreviations and Acronyms" section is incomplete, including abbreviations that appear in the abstract.

Author Response

# Reviewer 2

The main objective of the review is to discuss the use of hiPSC-derived cardiomyocyte models for drug screening and the identification of genetic and epigenetic variants, which begins in topic 4. However, until then, the authors provide an extensive and detailed review of other basic processes such as "CTRTOX Pathophysiology and Mechanisms" and "Recent Advancement in Therapeutic Strategies in the Treatment of CTRTOX", accompanied by tables and figures.

The main content of the manuscript is interesting and well detailed, but the other sections with excessive content impair the reader's experience. In addition, the authors need to review all the tables (sent in image form): the sentences are unformatted with centered texts at some points (such as in the therapeutic strategies column in Table 2). Sentences with periods appear at various times.

Several issues need to be reviewed:

Comment 1: The title needs modifications, in the current version it has ; and an acronym (hiPSC);

Response 1: Thank you for your valuable comments. We have removed the acronym from the title. Please refer to lines 2-4 in the revised manuscript.

Comment 2: Lines 46-48: "Furthermore, recent negative clinical trials have stymied prophylactic strategies to avert CTRTOX, whose devastating cardiovascular risks persist for months to years after completion of chemotherapy." This sentence needs a reference to clinical trials.

Response 2: Thank you for your valuable suggestion. While organizing the manuscript, we revised the statement mentioned above to read, “Furthermore, recent clinical trials have highlighted challenges associated with conventional chemotherapy strategies to avert CTRTOX and suggested alternative chemotherapy choices for cancer patients.” We have also cited the corresponding studies. Please refer to lines 45-47 in the revised manuscript.

Comment 3: Line 98: "demonstrated that (iPSCs) can be derived". It is necessary to remove the parentheses.

Response 3: Thank you for your valuable comments we have removed the parentheses. Please refer to line 99 in the revised manuscript.

Comment 4: Line 124: "human induced pluripotent stem cell-derived CMs (h-iPSC-CMs)". Throughout the manuscript, the authors do not use the hyphen. Furthermore, to talk about iPSC-CMs in other organisms (in line 106) the acronym would need to be defined.

Response 4: Thank you for your valuable comments. To keep the acronym consistent across the manuscript, we have removed the hyphen and modified it to hiPSCs-CMs. Please refer to line 124 in the revised manuscript. As suggested, induced pluripotent stem cell-derived CMs have been defined in the revised manuscript. Please refer to lines 107-108.

Comment 5: "cancer treatment related cardiotoxicity" can be used in the figure legend, but does not need to be in the image

Response 5: Thank you for your valuable comments. We have revised the figure accordingly. Please refer to the updated Figure 1 in the revised manuscript.

Comment 6: What does S. No. mean in Table 1? Why does DOX appear with four numbers in table 3

Response 6: Thank you for your observations. We have extensively revised all the tables. Please refer to the updated tables in the revised manuscript.

Comment 7: Line 172, 175: "hydrogen peroxide (H2O2) " "H2O2 production [61]. H2O2". The number 2 needs to be subscripted

Response 7: Thank you for your valuable comments as suggested the numbers are subscripted in the revised manuscript. Please refer to lines 174, 177 in the revised manuscript.

Comment 8: The authors use et al. several times in the manuscript, but 3 times they use "and colleagues". Line 245: " Kucerova and colleagues"; Line 574: " Imbaby and colleagues"; Line 782: " Tomoya and colleagues". This needs to be standardized.

Response 8: Thank you for your valuable comments. To keep it consistent throughout the manuscript, we replaced colleagues with et al. Please refer to line 248, 546, and 786 in the revised manuscript.

Comment 9: Several subitems have: as in "2.1.4. Idarubicin:", "2.2.1. Trastuzumab:", and "3.2. Delivery Strategies and Targeted Therapies:". This needs to be revised.

Response 9: Thank you for your valuable comments. As suggested, we have standardized the subitems throughout the manuscript. Please refer to the revised manuscript.

Comment 10: In section "2.2." the authors put the drug name with abbreviation in the title "2.2.2. Lapatinib (LAP):" while in other compounds not "2.2.3. Sunitinib". It is necessary to standardize.

Response 10: Thank you for your valuable comments. We have standardized the terminology in our revised manuscript and removed the abbreviation for Lapatinib from the title.

Comment 11: It would be interesting to summarize the information in "Table 1. Evidence of an effect of chemotherapy-induced cardiotoxicity." Of the 33 studies, only 7 (21%) involved patients, 11 involved cells (33%), which is precisely the focus of the review (using cell models for drug screening and the identification of genetic and epigenetic variants).

Response 11: Thank you for your valuable comments. We agree with the suggestion and have added that Table 1 summarizes the effects and mechanisms of the distinct chemotherapy strategies from the preclinical and clinical studies (7 (21%) involved patients, 11 involved cell models (33%) in our revised manuscript. Please refer to lines 135-137 in the revised manuscript.

Comment 12: In Figure 2, the authors use the citation number in the text. Since the image should be representative of the manuscript text, the authors should remove the references or cite them by name. This is different from a citation in a table, for example. In addition, numbers could be replaced by topic symbols.

Response 12: Based on the comment, we have removed citation numbers from the figure. Please refer to the revised Figure 2 in the manuscript. Again, we highly appreciate the reviewer’s suggestion, which has helped us enhance the clarity and rigor of our manuscript.

Comment 13: The tables are as images, so the resolution is not satisfactory.

Response 13: Thank you for your valuable feedback. We have revised all the tables to increase the font sizes and enhance readability. The updated tables are now inserted into the manuscript in a tabular format. Please refer to the revised manuscript.

Comment 14: In some sentences of Table 3, periods appear at the end. Authors should standardize this, including justified alignment. The table seems very disfigured and without resolution. Again, another table presented in the form of a figure. The text on the last line is in a different position than the others. In addition, two sentences have periods. The table needs to be reviewed.

Response 14: Thank you for bringing this to our attention. Based on the comment, we have revised all the tables to increase the font sizes, to enhance readability. Please refer to the updated tables in the revised manuscript.

Comment 15: The authors use "1. Introduction", but in the following sections "2.0. CTRTOX Pathophysiology and Mechanisms" and "4.0. Application of Human iPSCs-Derived Cardiomyocytes to Study CTRTOX" for example. The topic numbers should be 2 and 4, 2.1 is a subtopic, and there is no subtopic 2.0.

Response 15: Thank you for your valuable comments. The above-mentioned sections are now revised to match the introduction in our revised manuscript.

Comment 16: Topic 6 "Conclusion and Perspectives" needs to be completely reformulated. This concluding topic needs to be short and should address the main conclusions and perspectives in the field. The authors address new information. If they deem it necessary, this information should appear in previous topics.

Response 16: Thank you for your valuable comments. As suggested, we have revised the conclusion section to be shorter. We think that the new information added to the manuscript will be valuable and would allow researchers to develop safer chemotherapy regimens for cancer patients, which may be beneficial in developing personalized cardioprotectants and their application in clinical practice. Henceforth, following the reviewer’s suggestion we have merged it into the previous section. Please referred to the revised manuscript.

Comment 17: The "Abbreviations and Acronyms" section is incomplete, including abbreviations that appear in the abstract.

Response 17: Thank you for bringing this to our attention. We have expanded our list to include abbreviations and acronyms from the manuscript, including those appearing in the abstract.

Round 2

Reviewer 2 Report

Comments and Suggestions for Authors The authors have made the necessary suggestions. However, some details need to be corrected.   1. Table 1. Names in column 1 should be centered as in Table 3.   2. In the applications box in Figure 3 there is "and" and "&". This needs to be standardized.   3. The topic "6.0. Conclusion" was different from the standardization of the others.   4. In Figure 4, the authors place A through F, but did not cite them in the text or in the legend. Furthermore, the images inside the box (D) and (E) also have subdivisions that were not explored. This needs to be corrected.

Author Response

Itemized responses to the reviewers:

All changes and additions in the revised review manuscript are indicated in red.

# Reviewer 2

The authors have made the necessary suggestions. However, some details need to be corrected.

Comment 1: Table 1. Names in column 1 should be centered as in Table 3.

Response 1: Thank you for your valuable comments. Based on the comment we have revised the table accordingly. Please refer to Table 1 in the revised manuscript.

Comment 2: In the applications box in Figure 3 there is "and" and "&". This needs to be standardized.

Response 2: Thank you for your valuable suggestion. Based on the comment, we have revised the figure accordingly. Please refer to the updated Figure 3 in the revised manuscript.

Comment 3: The topic "6.0. Conclusion" was different from the standardization of the others.

Response 3: Thank you for your valuable comments. The conclusion section has now been revised to match the other sections. Please refer to the revised manuscript.

Comment 4: In Figure 4, the authors place A through F, but did not cite them in the text or in the legend. Furthermore, the images inside the box (D) and (E) also have subdivisions that were not explored. This needs to be corrected.

Response 4: Thank you for bringing this to our attention. Based on the comment, we have revised the figure accordingly. Please refer to the updated Figure 4 in the revised manuscript.
